# Structural insights into the mechanism of GTP initiation of microtubule assembly

Ju Zhou[1,2,3,5,7], Anhui Wang[4,7], Yinlong Song[2,6], Nan Liu[1,2,3], Jia Wang [1,2,3], Yan Li[4], Xin Liang [2], Guohui Li [4], Huiying Chu [4] ✉ & Hong-Wei Wang [1,2,3] ✉

In eukaryotes, the dynamic assembly of microtubules (MT) plays an important role in numerous cellular processes. The underlying mechanism of GTP triggering MT assembly is still unknown. Here, we present cryo-EM structures of tubulin heterodimer at their GTP- and GDP-bound states, intermediate assembly states of GTP-tubulin, and final assembly stages of MT. Both GTP- and GDP-tubulin heterodimers adopt similar curved conformations with subtle flexibility differences. In head-to-tail oligomers of tubulin heterodimers, the inter-dimer interface of GDP-tubulin exhibits greater flexibility, particularly in tangential bending. Cryo-EM of the intermediate assembly states reveals two types of tubulin lateral contacts, "Tube-bond" and "MT-bond". Further, molecular dynamics (MD) simulations show that GTP triggers lateral contact formation in MT assembly in multiple sequential steps, gradually straightening the curved tubulin heterodimers. Therefore, we propose a flexible model of GTP-initiated MT assembly, including the formation of longitudinal and lateral contacts, to explain the nucleation and assembly of MT.

Microtubules (MTs) are important components of the cytoskeleton networks in eukaryotic cells, responsible for cell shape determination, intracellular transport, cytoplasmic organization and cell division[1]. Playing these key roles in cellular activities, MTs have long been investigated as the drug target for cancer treatment[2–6]. The anti-cancer drugs function by either inhibiting MT polymerization, exemplified by the *Vinca* alkaloids, cryptophycins and colchicine, or antagonizing microtubule dynamics, like taxol and epothilones[7,8]. Additionally, there are a number of mutations in the tubulin gene that can result in complex cortical malformations known as tubulinopathies, which may be caused by a variety of factors including protein folding, α/β tubulin heterodimerization, or subsequent integration into growing microtubule polymers[9,10]. Therefore, the investigation of the molecular mechanism of MT polymerization is essential with both biological and pharmaceutical relevance.

MTs are hollow and polar tubular structure in vivo, mostly containing thirteen protofilaments (pfs)[11,12]. A pf is formed by head-to-tail stacking of αβ-tubulin heterodimers[13]. Each tubulin monomer has a nucleotide binding site in its N-terminal domain[14]. The GTP bound to α-tubulin is buried in the α-, β-tubulin intra-dimer interface and is non-exchangeable and non-hydrolysable, whereas the nucleotide bound to β-tubulin is exposed to solvent and exchangeable[15,16]. A tubulin heterodimer with GTP-bound to β-tubulin is termed as GTP-tubulin, while that with GDP bound to β-tubulin is termed as GDP-tubulin. The β-tubulin-bound GTP can be hydrolyzed upon the incorporation of the GTP-tubulin into microtubule lattice, which plays important roles in MT dynamic instability[17]. When GTP-tubulin accumulates at the growing end and forms a "GTP cap", the MT is stabilized and continues polymerizing. Once the "GTP cap" is lost, MT catastrophe occurs.

[1]State Key Laboratory of Membrane Biology, Tsinghua University, Beijing 100084, China. [2]Tsinghua-Peking Joint Center for Life Sciences, School of Life Sciences, Tsinghua University, Beijing 100084, China. [3]Beijing Frontier Research Center for Biological Structures, Tsinghua University, Beijing 100084, China. [4]Laboratory of Molecular Modeling and Design, State Key Laboratory of Molecular Reaction Dynamics, Dalian Institute of Chemical Physics, Chinese Academy of Science, 457 Zhongshan Road, Dalian 116023, China. [5]Present address: University of California Berkeley, Berkeley, CA, USA. [6]Present address: Cell Biology, Neurobiology and Biophysics, Department of Biology, Faculty of Science, Utrecht University, Utrecht, Netherlands. [7]These authors contributed equally: Ju Zhou, Anhui Wang. ✉e-mail: chuhy2009@dicp.ac.cn; hongweiwang@tsinghua.edu.cn

The nucleotide state of tubulin dimer has a deterministic effect on the MT assembly cycle, especially in the early stages of MT polymerization. However, whether the GTP-binding and hydrolysis affect the polymerization state of tubulin by a direct conformational change of the protein remains a controversy. In early studies, people observed that the ends of growing MTs have straight pfs and those of shrinking MTs have curling or coiled pfs[18,19], proposing that GTP-tubulin is straight while GDP-tubulin is curved, and GTP triggers the conformational transition of curved dimer to straight dimer. Nevertheless, this opinion was challenged by the theory of "lattice model", based on an observation of the long growing ends to be more gently curved rather than previously reported straight[20–22]. The "lattice model" proposes that the lateral interaction of tubulins during the cylindrical closure of MT straightens the curved dimer. Cryo-EM analysis of the assembly of GDP-tubulin mimicking the shrinking MT ends demonstrated an intrinsic bending of ~10 degrees within the dimer, while that of an ordered assembly of tubulin bound by a non-hydrolysable GTP analog, GMPCPP, mimicking the growing MT ends showed an intrinsic curvature of ~5 degrees within the dimer[23–25]. Since the first atomic structure of αβ-tubulin dimer was solved by electron crystallography in the "zinc-sheet" assembly[14], many atomic models of tubulin have been solved by X-ray crystallography to reveal the unpolymerized structure of tubulin dimers in different nucleotide states bound with various depolymerizing cofactors and/or tubulin binding proteins[26–32]. In most of the structures, both GDP- and GTP-tubulin adopted similar curved conformation with a curvature of nearly 10 degrees within the dimer, despite of its binding cofactors. Crystal structures showed that GTP at the β-tubulin induced a small structural change with the flip of T5 loop close to the inter-dimer interface[31,32]. A major drawback of the crystal structures, however, is the usage of MT depolymerizers to block the tubulin assembly process[26,28–31,33]. The effect of MT depolymerizers as well as the crystal packing on the conformation of tubulin cannot be ruled out when analyzing the MT assembly mechanism.

Additionally, MD simulations have been widely utilized to study potential mechanisms of dynamic instability in MT, including tubulin-tubulin lateral and longitudinal bonds[34,35] as well as intra- and inter-dimer flexibility[36–39]. A combined model assuming that the allosteric effects retain the flexibility of intra- or inter-dimer interface dominated by lattice induced effects was developed[36,37,40,41]. The simulation results based on the combined model suggest that the nucleotide induces the intrinsic flexibility difference rather than the large conformational changes in tubulin, which are responsible for the assembly of MTs. More recent studies, including simulations of MD and modeling based on phenomenology, have demonstrated that activation energy barrier of lateral interactions is a significant factor in microtubule assembly[42,43]. While many MD simulations start with the tubulin structure derived from the cryo-EM structure of MT, this may not fully reflect tubulin conformational changes in solution.

The enigma of nucleotide's allosteric effect on tubulin is mainly due to the lack of high-resolution structures of tubulin dimer in its native states during the assembly process. In this work, we use single particle cryo-EM to analyze the structures of tubulin heterodimer in the GDP- and GMPCPP-bound solution states without any other cofactors or depolymerizing reagents at resolutions ranging from 3.5 Å to 3.9 Å, revealing the authentic effect of nucleotide on tubulin heterodimer. We have also solved the intermediate structure of GTPγS-tubulin mimicking the early stage of MT assembly at a 6.8 Å resolution and the well-assembled GTPγS-MT at 4.3 Å resolution, respectively. Based on the structural analysis and accompanying MD simulations, we have put forth a flexible model describing the straightening process of the tubulin heterodimer during assembly. Our model suggests that the binding of GTP plays a crucial role in reducing the intrinsic flexibility of the interdimer, particularly in terms of tangential bending. This reduction in tangential bending flexibility enhances the stability of lateral interactions during the formation of the microtubule lattice, consequently promoting efficient microtubule assembly.

## Results

### Structures of GDP-tubulin and GMPCPP-tubulin heterodimer in solution

In order to reveal the structure of tubulin heterodimer in solution, single particle cryo-EM is employed to analyze tubulin bound to GDP or GMPCPP, respectively (Methods, Supplementary Figs. 1 and 2). The porcine tubulin is incubated with GDP or GMPCPP on ice, then the dimeric fraction is isolated by gel filtration chromatography. Afterwards, we have collected and analyzed the GDP-tubulin and GMPCPP-tubulin cryo-EM datasets separately. As a result, we have classified and identified two major conformations for GDP-tubulin dataset as GDP-1 (2/3 of the population) and GDP-2 (1/3 of the population) conformers at 3.6 Å and 3.9 Å resolutions, respectively (Supplementary Fig. 1c, d). Comparatively, only one major conformation has been found from the GMPCPP-tubulin dataset and a 3D reconstruction is obtained at 3.5 Å resolution (Supplementary Fig. 2d, e). All the 3D EM maps are good enough for us to build atomic models of these different conformers that represent the authentic tubulin heterodimer states in solution (Supplementary Fig. 3). Compared with those previously reported tubulin heterodimer structures, all of them have a similar curvature of ~12° (Supplementary Fig. 4a, b). When superimposing these tubulin intra-dimer structures on α-tubulin, we could observe very small variation on β-tubulin structure (Supplementary Fig. 4a). Even though the two tubulin dimers are derived from the same tubulin tetramer crystal structure, they also show minor differences (Supplementary Fig. 4a). Thus, we consider these minor structural vibrations to not indicate significant structural differences, but instead indicate a minor structural flexibility within the intra-dimer interface.

With the atomic models of tubulin dimers in solution, we have performed structural comparison among these different conformers of tubulin in GDP (GDP-1 and GDP-2) and GMPCPP nucleotide states. The conformation of GDP-1 state, as the major population of GDP-tubulin, shows a high similarity with that of GMPCPP state, with displacements of most Cα atoms <1 Å (Fig. 1a, b). In contrast, the GDP-2 state shows displacements nearly 2 Å for most regions in β-tubulin, especially in the H2-S3 and H1-S2 loops, indicative of relatively larger structural variation from GMPCPP-tubulin (Fig. 1c, d). Furthermore, the EM density of the M- and H2-S3 loops of GDP-states have poorer quality than those of the GMPCPP-tubulin (Fig. 1e–g), also indicating a more flexible conformation of the GDP-tubulin than that of the GMCPP-tubulin.

Additionally, the Root-Mean-Square Deviation (RMSD) analysis between GDP-1 and GDP-2 structures reveals that the structure around the β-phosphate of GDP is more divergent than those approximate to the guanine nucleobase (Supplementary Fig. 1e). Furthermore, when comparing the tubulin structures of GMPCPP and GDP-2 state, the region near the γ-phosphate shows a bigger difference than that surrounding the guanine nucleobase (Fig. 1d). According to the RMSD value distribution, the N-terminal domain of the β-tubulin is a more divergent region. In this region, the H2-S3 loop is positioned nearby the phosphate side of the nucleotide, hinting that the the γ-phosphate is positioned to stabilize the GTP binding pocket and its surrounding regions, resulting in a more rigid conformation of the tubulin heterodimer (Fig. 1d). In addition, the Root-Mean-Square Fluctuation (RMSF) analysis of MD simulations of tubulin heterodimer between GDP and GTP states also shows that the GDP-bound state has much larger fluctuations in the N-terminal 1–120 amino acids near the phosphate side, especially those from H2 to H3 helix (Supplementary Fig. 5). Therefore, GTP analogs are able to stabilize the nucleotide binding domain and such a local structural stabilization propagates through the β-tubulin and allosterically induces the GTP-tubulin into a more homogeneous conformation.

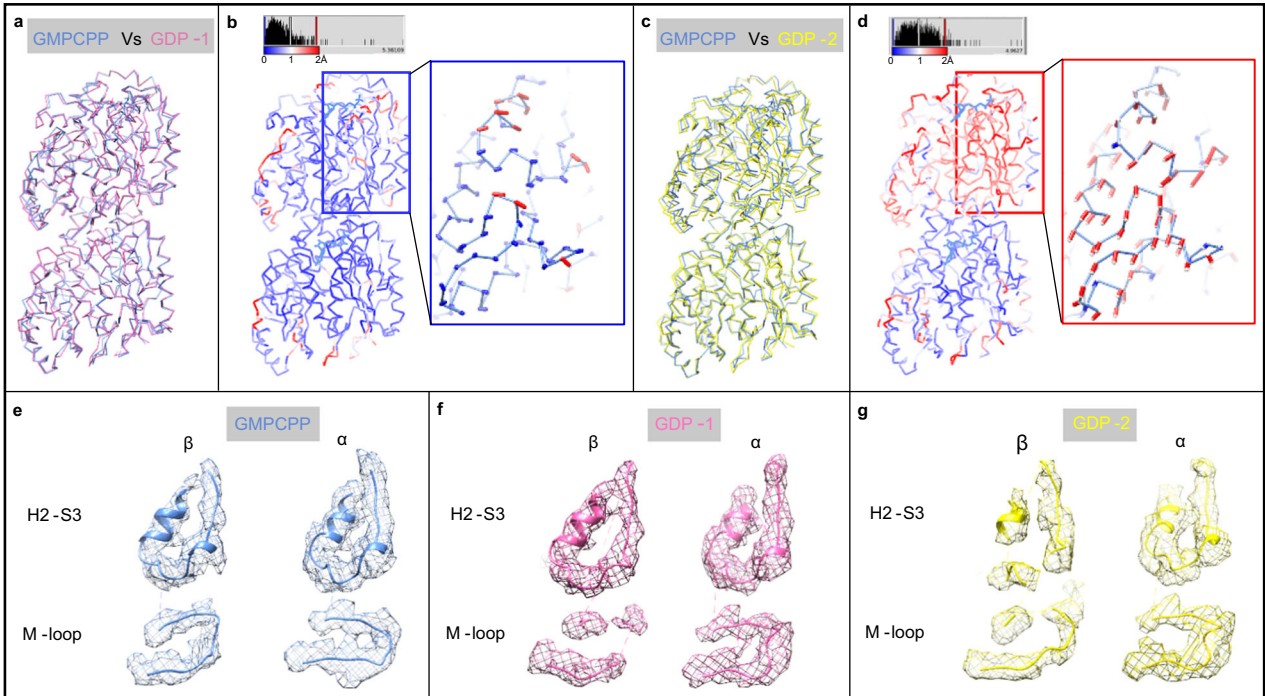

**Fig. 1 | Comparison of GMPCPP- and GDP-tubulin heterodimer structures in solution. a** Cα-trace superimposition of the two tubulin heterodimer models between GMPCPP (cornflower blue) and GDP-1 (hot pink) state, aligned on the α-tubulin (same for **b**–**d**). **b** Cα-atoms-RMSD between the two models shown in **a**, with deviations colored from blue to red. The chain-trace being displayed corresponds to the GMPCPP-tubulin model. Displacement vectors of Cα-chain of H1-S2 and H2-S3 loops in β-tubulin are represented by heavy lines in the inset. The vector >1 Å is colored in red and <1 Å is colored in blue (same for **d**). **c** Cα-trace superimposition of the two tubulin heterodimers between GMPCPP (cornflower blue) and GDP-2 state (yellow). **d** Cα-atoms-RMSD between the two models shown in **c**. The inset shows the displacement vectors of H1-S2 and H2-S3 loops. **e**–**g** The density of cryo-EM maps (mesh) and corresponding models of H2-S3 loop and M-loop. β-tubulin is shown in the left panel and α-tubulin in the right panel. **e** GMPCPP state. **f** GDP-1 state. **g** GDP-2 state. Electron densities are contoured at 3.06 σ.

## The inter-dimer interface of GDP-tubulin and GTP-tubulin tetramer

Apart from the tubulin heterodimers, we are able to identify certain amount of particle images of head-to-tail tubulin tetramer and hexamer in the cryo-EM micrographs, probably due to a spontaneous self-assembly of tubulin in solution. 2D analysis of the tetramers have revealed different inter-dimer curvatures of tubulin existing in both GDP and GMPCPP states (Supplementary Figs. 6 and 7). Unlike the fixed structure of tetramer solved by X-ray crystallography[26,27,31], the tubulin tetramers in solution adopt a series of different conformations (Supplementary Fig. 4c). Even though we could not get high resolution structures of tubulin tetramer due to preferential orientation problem, the density maps are good enough for us to do rigid-body fitting with our solved tubulin dimer models. It is obvious that both GDP- and GMPCPP-tubulin have demonstrated a greater variation of their inter-dimer interfaces than their intra-dimer interfaces (Figs. 1a–d and 2a).

To assess and better understand whether the nucleotide plays an important role in the inter-dimer interaction, we have decomposed the overall rotation between the two dimers within a tetramer into bending angles of three perpendicular directions: radial, tangential and twist (Fig. 2b). In these three directions, the variation range of radial bending angle, ranging from 20° to 44° in GDP-tubulin and 23° to 41° in GMPCPP-tubulin (Fig. 2c, f and Supplementary Table 1), is much larger than that of tangential bending and twist angles, both of which are no more than 10° (Fig. 2d, e, g, h and Supplementary Table 1). Building upon the bending angles in each direction (Supplementary Table 1), we conducted a further comparative analysis of the angle deviation in the inter-dimer interfaces between the GDP and GMPCPP states. Specifically, we generated new datasets using bootstrapping framework and calculated the weighted standard deviations of the radial bending, tangential bending, and twist angles for the GDP-bound tubulin

tetramers. These values were found to be 7.7°, 2.2°, and 3.2°, respectively. In contrast, the corresponding standard deviations for the GMPCPP states were 6.4°, 1.5°, and 0.7°, respectively. The statistical significance of these differences was confirmed through the application of the T-test (Fig. 2i–k and Supplementary Table 1). Based on these results, it can be observed that the GDP-tetramer exhibits greater fluctuations in all directions compared to the GMPCPP state. This analysis provides valuable insights into the dynamic behavior of the tubulin tetramers and supports the notion that the presence of GDP influences the flexibility of the structures.

For more dynamic comparison, we conducted further 3D variability analysis (3DVA) by combining three classes of GDP- and GMPCPP-tubulin datasets separately for refinement. The 3DVA confirms that both GDP- and GMPCPP-tubulin tetramer are flexible, swinging within a certain range in solution (Supplementary Movies 1–4). Notably, the inter-dimer interface of GDP- and GMPCPP-tubulin tetramers displays considerable structural variations in the radial bending direction (Supplementary Movies 1 and 3). However, a particularly interesting observation is that significant variations in tangential bending occur around the inter-dimer interface of the GDP tubulin tetramer, while such variations are absent in the GMPCPP-tubulin tetramer (Supplementary Movies 2 and 4). This finding indicates that the GDP-tubulin tetramer is more flexible than its GMPCPP counterpart, specifically in terms of tangential bending. This flexibility in tangential bending is particularly important for the formation of lateral contacts, which play a critical role in the overall assembly process.

In conclusion, tubulin tetramer does not adopt a fixed conformation in solution and the fact that GDP-tubulin tetramer is more flexible than GMPCPP-tubulin indicates that the flexibility of the inter-dimer interface is regulated by the nucleotide bound to β-tubulin. Compared to GMPCPP state, GDP-tubulin tetramer displays greater

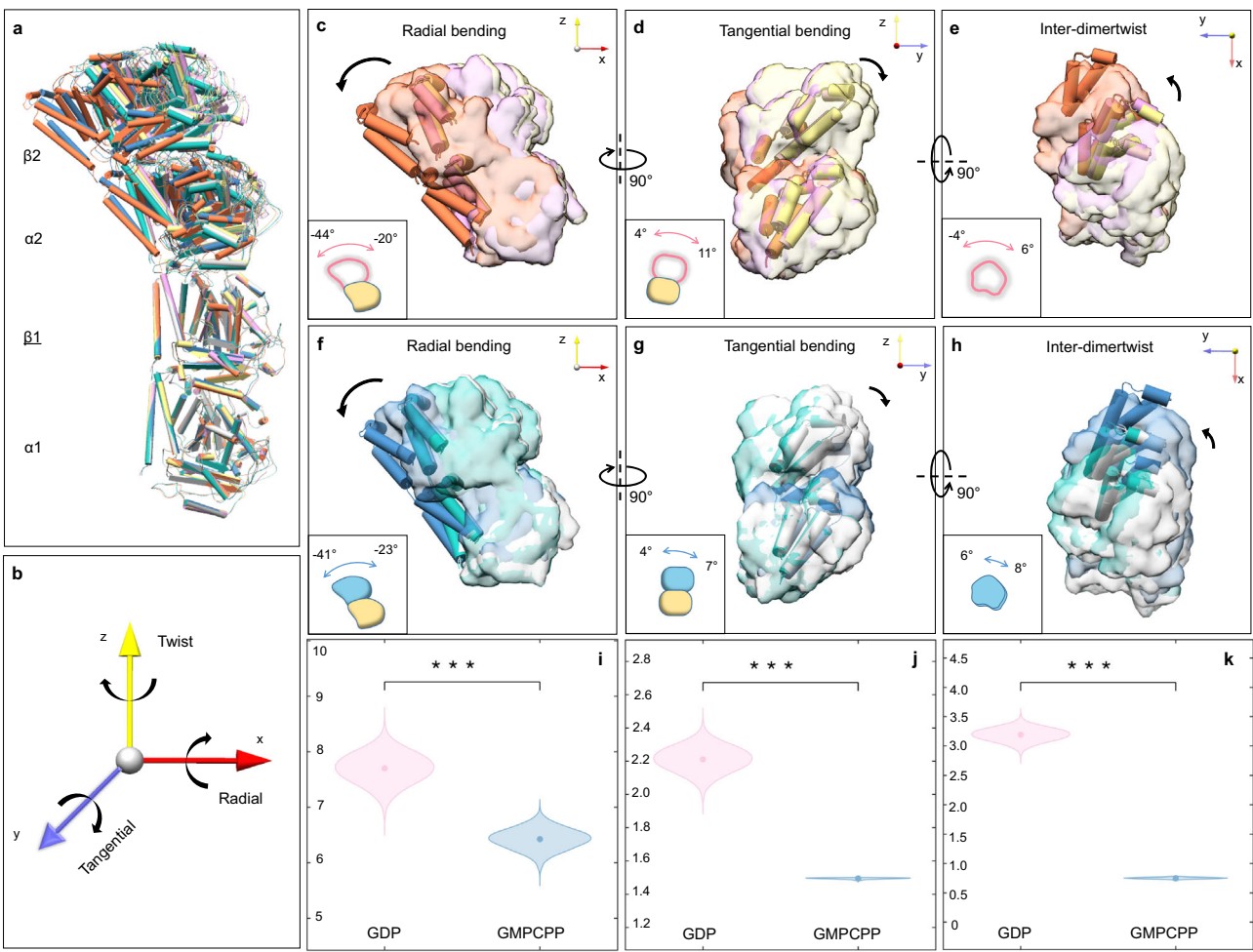

**Fig. 2 | The curvature fluctuation around the inter-dimer interfaces of tubulin tetramer in solution. a** Overview of three major conformations of GDP- and GMPCPP-tubulin tetramers. All models are aligned together using β1-tubulin as reference. GDP-tubulin tetramers are colored with coral, plum and khaki, respectively. GMPCPP-tubulin tetramers are colored with steel blue, light sea green and light gray, respectively. **b** The coordinate system is defined to describe the bending angles of α2, β2-tubulin relative to α1, β1-tubulin. And the quantitative values of the angles are decomposed in the x (radial bending), y (tangential bending) and z (twist) axes. **c–h** Bending of inter-dimer in radial, tangential and twist directions. The upper heterodimer (α2, β2-tubulin) shown in (**a**) of GDP-state (**c–e**) and GMPCPP-state (**f–h**). The black arrow indicates the direction of bending. A simplified cartoon model and its bending angle range are displayed in the left-bottom corner. H11 and H12 helices are represented as cylinders, the N and I domain are shown as surfaces. **i–k** Violin plots representing the standard deviation of bending angles are shown for GDP and GMPCPP states, obtained from bootstrapping datasets for a two sample $T$-test. The circles in each related violin plot indicate the mean value of the standard deviation angle. The vertical axis represents the standard deviation of bending angles, while the horizontal axis corresponds to different nucleotide states. The two sided $T$-test assumed the standard deviation angles of GDP and GMPCPP are in the same distributions with equal means in 1 dimensional space with the 95% confidence interval. Statistically significant results from the two sided $T$-tests are denoted as follows: *$p$-value < 0.05; **$p$-value < 0.01; ***$p$-value < 0.001. In all three tests, the hypotheses are rejected. **i** Violin plot displaying the distribution of standard deviation in radial bending angles. The $p$-value is $1*10^{-11}$ and effect size is 5.38. **j** Violin plot displaying the distribution of standard deviation in tangential bending angles. The $p$-value is $1*10^{-12}$ and effect size is 13.13. **k** Violin plot displaying the distribution of standard deviation in twist angles. The $p$-value is $1*10^{-12}$ and effect size is 30.07.

flexibility, especially in the direction of tangential bending, which is crucial for lateral contact formation, and the flexibility could make it difficult to interact with neighboring tubulin molecules laterally. Due to the small size of the particle stack and more heterogeneous and complicated conformations of the tubulin hexamer particles, we did not perform 3D reconstruction of these particles for further study (Supplementary Fig. 8).

## Capture of MT assembly intermediates from *Drosophila* S2 tubulin

Since the structures of tubulin in solution have demonstrated a rather weak effect of nucleotide on the tubulin intra-dimer curvature, we seek to understand the conformational change of tubulin during the MT assembly process by examining the structure of tubulin in a MT assembly intermediate that bridges the gap between the initial and final state of MT assembly. In previous works, an inside-out "GMPCPP-Tube" assembled from GMPCPP-bovine tubulin was considered to mimic the transient intermediates in the growing MT end[23–25,44]. In an attempt to study the early MT assembly process using fruit fly tubulin endogenously purified from *Drosophila* S2 cells, we have found a similar intermediate structure in the presence of another GTP-nonhydrolyzable analog GTPγS. Notably, unlike the "GMPCPP-Tube" made of bovine tubulin, the S2-GTPγS-tubulin assembly intermediate can form under physiological Mg²⁺ concentrations, thus closer to the native MT assembly state. Different from bovine tubulin, S2 GTPγS-tubulin is able to nucleate spontaneously without high glycerol concentration and often shows a long helical ribbon attached to a growing MT's end (Fig. 3a). The helical ribbon is similar to the reported self-assemblies of GMPCPP-tubulin[24]. Some ribbons may grow into a tube-like structure very similar to the previously reported "GMPCPP-Tube", therefore here termed as "GTPγS-Tube", which can co-exist with GTPγS-MTs in solutions of 2–5 mM Mg²⁺ concentration

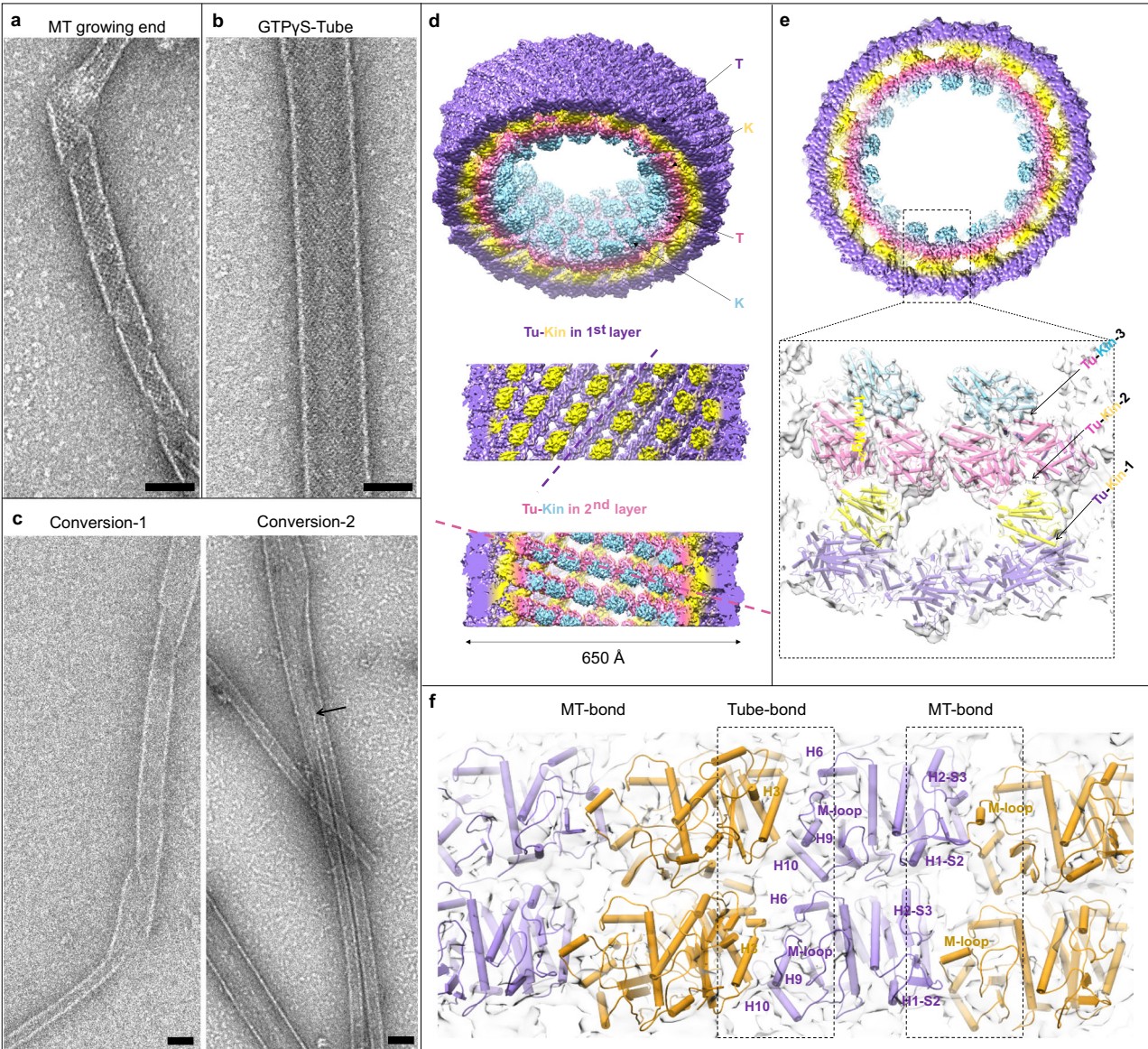

**Fig. 3 | Negative-staining EM micrographs and cryo-EM structure of the MT assembly intermediate. a–c** EM micrographs of negatively stained specimens. Scale bar: 50 nm. **a** Helical ribbons appear at the growing end of GTPγS-MT at 28 °C with 2 mM Mg²⁺. **b** Only GTPγS-Tube exists in the condition of 10 mM Mg²⁺. **c** GTPγS-Tubes convert into MT directly in two different ways. Left (Conversion-1): One Tube converts into one MT. Right (Conversion-2): One Tube splits into two MTs. The branch point is pointed out by the black arrow. Three independent replications of negative staining were conducted. **d–f** Cryo-EM structure of GTPγS-Tube-KMD complex. **d** The density map of GTPγS-Tube-KMD complex. Top: Slightly tilted top view of the whole complex. Middle: Side view of the outer tubulin (medium purple) and kinesin (yellow). Bottom: Side view of the inner tubulin (hot pink) and kinesin (cyan). The dashed line indicates the direction of tubulin heterodimer arrangement. **e** A cross section of the whole map. Atomic models of tubulin and kinesin are fitted in the density map, and three tubulin-kinesin interfaces are labeled (inset). **f** The density map and corresponding models of four adjacent tubulin heterodimers in the outer layer of GTPγS-Tube. "Tube-bond" and "MT-bond" interfaces are indicated by the black box with dashed line. Secondary structures engaging in these interfaces are labeled.

(Supplementary Fig. 9a, b). Further increasing the Mg²⁺ concentration to 10 mM favored the formation of only GTPγS-Tube (Fig. 3b). These Tubes are centrifuged and resuspended in BRB80 buffer with 1 mM Mg²⁺ concentration, followed by negative-staining EM analysis. We have observed conversion process from the Tube to MT, showing structure intermediates with one end as a wider Tube and the other end as a narrower MT (Fig. 3c left and Supplementary Fig. 9c–f). We have found at least two means of the conversion in the negative-staining EM micrographs (Fig. 3c right and Supplementary Fig. 9c–f). In the first means, the Tube is unwound first and then spirals into a MT, and thus one Tube becomes one MT (Fig. 3c and Supplementary Fig. 9c, d). In the other means, one Tube can be straightforwardly divided into two MTs, and the branching point is clearly discernible in

the micrograph (Fig. 3c and Supplementary Fig. 9e, f). In conclusion, the GTPγS-ribbon likely represents the tubulin assembly intermediate in MT growth.

### Cryo-EM reconstruction of GTPγS-Tube

We decided to gain more structural information about the tubulin interactions in the GTPγS-ribbon assembly by performing cryo-EM reconstruction of the GTPγS-Tube with helical symmetry. But unfortunately, we are not successful to obtain a 3D reconstruction map of GTPγS-Tube with enough structural details by itself. To improve the quality of the helical order and distinguish the α- and β-tubulin, we have incubated the S2-GTPγS-tubulin with kinesin-1's motor domain (KMD) in the condition for GTPγS-Tube assembly and have got a more ordered

helical assembly decorated with KMD on each tubulin heterodimer. Such kind of sample has enabled us to solve the cryo-EM structure of S2-GTPγS-Tube at 6.8 Å resolution (Supplementary Fig. 10).

The 3D reconstruction of GTPγS-Tube-KMD is of good enough quality to reveal important interactions among tubulin dimers in the helical assembly, which consists two layers of tubulin-KMD complex, the outer tubulin and KMD forming the first layer and the inner tubulin and KMD forming the second layer (Fig. 3d and Supplementary Movie 5). There are three interaction interfaces between tubulin and KMD, named Tu-Kin-1, Tu-Kin-2 and Tu-Kin-3, respectively (Fig. 3e and Supplementary Fig. 11a, b). The Tu-Kin-1 and Tu-Kin-3 interfaces are highly conserved electrostatic interaction in the Kinsin-1 family with MTs, in which the α−4 helix and loop 11 of kinesin interact with the H8-S7 loop of β-tubulin, and the H11′, H12 helices of α-tubulin and the loop 8 of kinesin interacts with the H12 helix of β-tubulin[45]. The Tu-Kin-2 interface is a different interface, surrounded by a few charged residues and several flexible loops (Supplementary Fig. 11c), involving the H1-S2, H2-S3, S8-H10 and H6-H7 loops located in the tubulin intra-dimer area to interact with the β0 and β1 strands of KMD. Another protein in kinesin family, kinesin-13s, also have been reported to have a second tubulin binding site, which is important for spindle morphogenesis and poleward chromosome movement during mitosis[46,47]. Whereas, the second tubulin-kinesin interaction interface of kinesin-13s is distinct from the Tu-Kin-2 interface of kinesin-1. The region outside of the MT interacts with kinesin-13 and the area within the MT lumen interacts with kinesin-1. At this point, it is unclear whether the interface we have discovered might play a role in vivo.

We are mostly interested in the structure of tubulin and their interactions in the Tube assembly in comparison with those in a fully assembled MT. We therefore use the same S2-GTPγS-tubulin to assemble MTs and obtain cryo-EM reconstruction of S2-GTPγS-MT decorated by KMD at a resolution of 4.3 Å (Supplementary Fig. 12). The structure and lattice interaction between tubulin heterodimers in the MTs are highly conserved between the S2 and porcine tubulins (Supplementary Fig. 12f, g). Within the 15-pf S2-GTPγS-MT, the adjacent protofilaments interact laterally with each other via a conserved interface by M-loop, H1-S2 and H2-S3 loop between two adjacent tubulin monomers, named as the "MT-bond" interface (Supplementary Fig. 12g). The Tube structure is overall very similar to the previous ~20 Å resolution reconstruction of the "GMPCPP-Tube"[23] but with much better defined structural details. In the Tube, two different types of lateral interaction interfaces are observed between neighboring curved protofilaments spiraling around the helical axis (Fig. 3d, f). One interface is the same as that in the MT lattice, the "MT-bond" interface, whereas the other interface is unique in the Tube lattice, therefore named as the "Tube-bond" interface. The "Tube-bond" is mainly sustained with the interaction among the M-loop, H10-S9 loop, H9-S8 loop, H3, H4, H6, H9 and H10 helices (Fig. 3f and Supplementary Movie 5).

### The conversion from "Tube-bond" to "MT-bond"

For better understanding of the kinetic and dynamic property of the two types of lateral interaction, we have performed MD simulations on the "MT-bond" and "Tube-bond" interfaces. Interestingly, these two interfaces have different binding free energies, with the "Tube-bond" being −26.23 kcal/mol and the "MT-bond" being −45.40 kcal/mol (Supplementary Table 2). Hence, the "MT-bond" interface is more stable than the "Tube-bond" interface, since its binding free energy is much stronger. While the electrostatic interaction contributes mostly to both interfaces, the major secondary structural elements and dominant residues involved are different (Fig. 4a, b). Consistent with those previously reported microtubule structural studies[48–51], the "MT-bond" is a highly conservative interface composed of M-loop, H1-S2 loop and H2-S3 loop. The α:R339 contributes the most to both "Tube-bond" (−8.67 kcal/mol) and "MT-bond" formation (−9.60 kcal/mol),

but interacts with α:E196 in the former and α:D160 in the latter interface. α:R214 forms a salt bridge with E113 in the "Tube-bond" interface, whereas this lateral interaction is disrupted by β:E330 in the "MT-bond" interface. Some residues in H1-S2 (β:N54) and H2-S3 loops (β:R88 and β:D90) are engaged in the "MT-bond" interface, which are not involved in the "Tube-bond" formation (Fig. 4a, b and Supplementary Table 2).

Combined with the structural data and EM observation mentioned above (Fig. 3c, f and Supplementary Fig. 9c–f), we infer that the "Tube-bond" may convert into "MT-bond" directly. To verify this idea, we have conducted additional structure-based MD simulations, which is widely used in recent studies of the assembly of ribosomes or protein filaments[52–55]. Here, we have defined the "MT-bond" configuration as a potential energy minima and used solution tubulin structures as initial states to explore the lateral interaction formation process and reveal key interactions between neighboring dimers that drive the process. In agreement with our experimental observation, the MD simulations demonstrate the initial formation of the "Tube-bond" followed by the conversion into the "MT-bond" (Fig. 4c–g and Supplementary Movie 6). This continuous process can be divided into three major steps, named "Tube-bond Formation", "Tube-bond Dissociation" and "MT-bond Formation". Here, we have listed some intermediate lateral interactions during these steps. The α-tubulins form "Tube-bond" via lateral interaction initially triggered by the electrostatic interaction between the R339 and five negative charged residues (E414, E417, E420, E155 and E196) (Fig. 4d and Supplementary Fig. 13a). Then D414 and E159 of β-tubulin form salt bridges with R308 and K338, respectively (Fig. 4e and Supplementary Fig. 13b). Later, these interactions break apart when two adjacent β-tubulins separate (Fig. 4f and Supplementary Fig. 13c). Instead, D160 and R123 of α-tubulin form salt bridges with R308 and E290 in the neighboring one, triggering the formation of "MT-bond" interface (Fig. 4g and Supplementary Fig. 13d). The simulations have demonstrated vividly a possible scenario during the MT assembly in which the "Tube-bond" forms fast in a kinetically favored event and later converts directly into the thermodynamically more stable "MT-bond" (Supplementary Movie 6).

### The conformational changes of tubulin heterodimer induced by lateral interactions

Thus far, we have obtained the tubulin heterodimer conformation at three stages before (dimer in solution), during (helical ribbon and Tube) and after (MT) the MT assembly. Accompanying to the process of MT assembly, the curvature of tubulin intra-dimer changes from ~12°, ~6°, to ~0°. In order to better describe the conformational change of αβ-tubulin heterodimer during the MT assembly process, we have superimposed all the structures using α-tubulin as the reference and focused on the structural changes of β-tubulin. The α-tubulin in all states shows quite similar structure and aligns well.

The comparison of the structures between the first two states shows that the β-tubulin undergoes a rotation around the intra-dimer interface (Fig. 5a, c). This rotation can be divided into the radial and tangential direction (Fig. 5a–d and Supplementary Table 3). The displacement in the tangential direction is larger than that in the radial direction (Fig. 5b, d), resulting a decreased intra-dimeric bending angle by ~8° and ~4° (Supplementary Table 3). According to the MD simulations, the major change of intra-dimer curvature occurs at nearly 2.5 μs, when the "Tube-bond" interface dissociates and "MT-bond" interface forms (Supplementary Fig. 14 and Supplementary Movie 6). It appears that the "Tube-bond" forms quite fast even before the intra-dimer interface changes. Therefore, the intra-dimer curvature change is mostly caused by the formation of "MT-bond". Structural comparison between tubulin dimer in the Tube and MT lattice displays another rotation of β-tubulin, decomposing into a small bending angle of ~2° in radial direction and a large twist angle of ~4° (Fig. 6a–d and Supplementary Table 3).

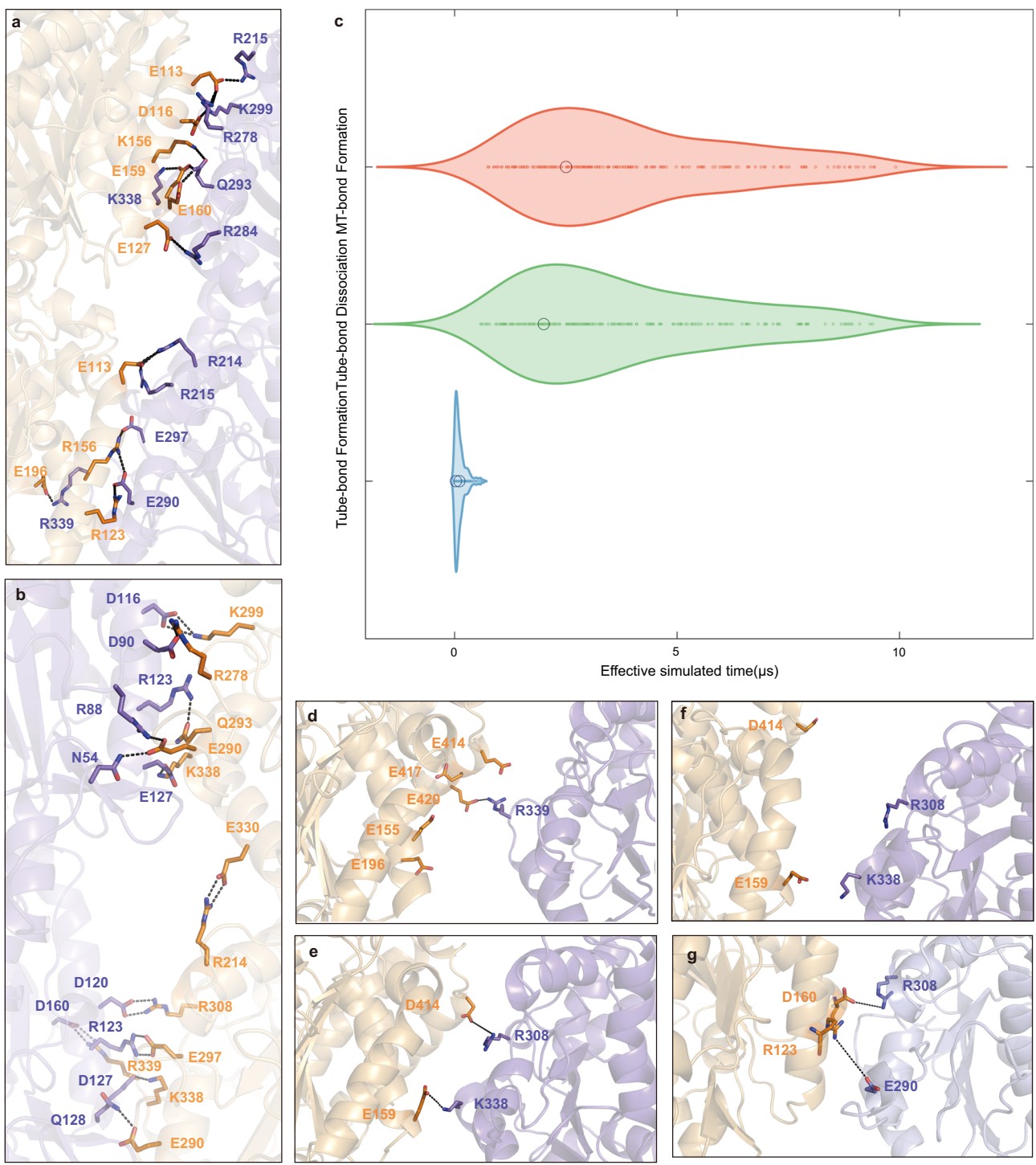

**Fig. 4 | The "Tube-to-MT" conversion revealed by MD simulations. a, b** Two snapshots of the MD simulations show different lateral interactions. The dominant amino acid residues are listed in the "Tube-bond" interface (**a**) and "MT-bond" interface (**b**). The tubulin model is depicted as a ribbon structure, with key residues represented as sticks. Dark orange and light purple indicate the adjacent tubulin subunits. **c** The violin plots demonstrate the distribution of effective simulated time in three main states of "Tube-bond Formation" (blue), "Tube-bond Dissociation" (green) and "MT-bond Formation" (red). **d–g** Snapshots of the MD simulations show the lateral interface between two tubulin monomers during the process of "Tube-bond Formation" (**d**, **e**), "Tube-bond Dissociation" (**f**) and "MT-bond Formation" (**g**). The dominant residues are shown as sticks.

Combined with our Tube and MT structure, one side of "MT-bond" formation brings a half of curvature change of tubulin intra-dimer and two sides of "MT-bond" formation leads to the fully straightened conformation.

The M-loop has played important roles in the lateral lattice formation. Once two tubulin heterodimers form a "MT-bond", the unstructured M-loop inserts into the complementary "lock" formed by

the adjacent H1-S2 and H2-S3 loops, causing the M-loop to fold into a short helix and become a stable structural element. Meanwhile, its surrounding regions such as the H6-H7 and S9-S10 loops have a displacement of about 10 Å (Figs. 5e and 6f). These structural rearrangements partially straighten the tubulin intra-dimer curvature. Once the other side of the tubulin heterodimer also forms a "MT-bond", the H1-S2 and H2-S3 loops moves closely to the neighbor M-loop with a

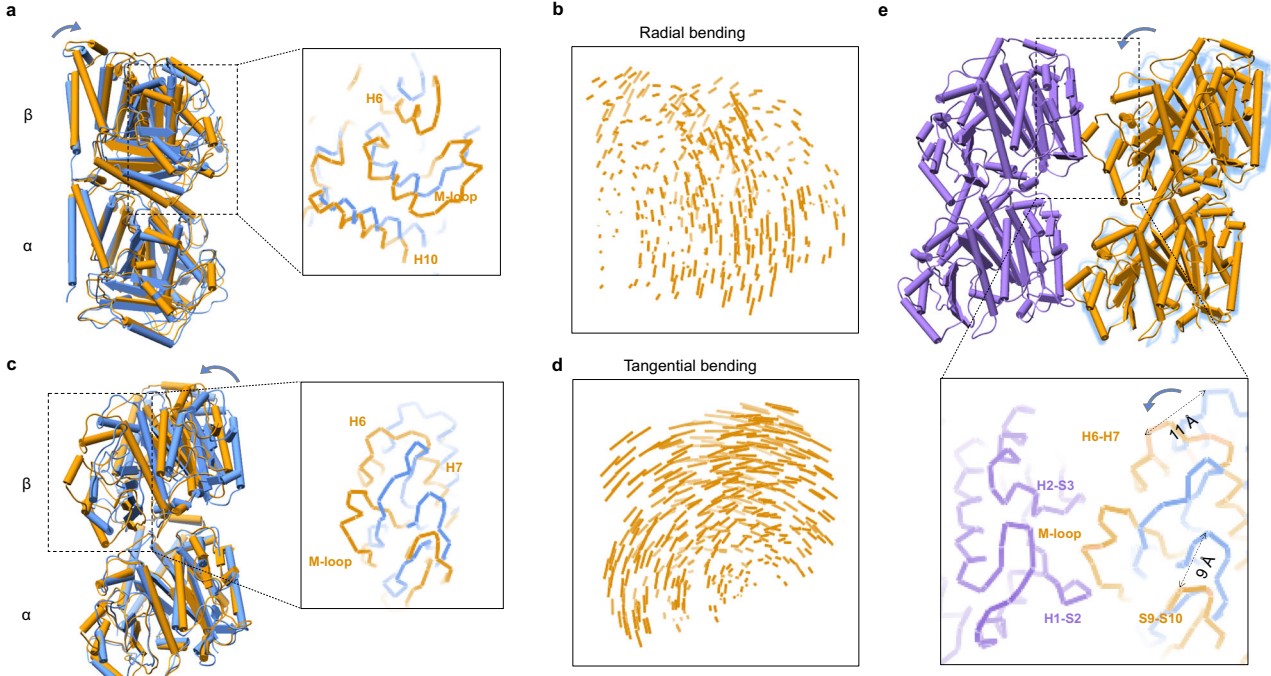

**Fig. 5 | M-loop gets ordered upon forming one side of the lateral contacts of "MT-bond".** **a–d** Structural comparison between the initial state of GMPCPP-tubulin heterodimer in solution (blue) and the intermediate state of tubulin heterodimer in the Tube lattice (orange). Models are superimposed on the α-tubulin. The direction of movement is marked by the blue arrow. **a** Side view. The inset shows the movements of M-loop and its surrounding regions (same for **c**). **b** Displacement vectors of β-tubulin Cα coordinates comparison between two tubulin heterodimers shown in (**a**). The length of the stick correlates with its displacement (same for **d**). **c, d** Structural changes (**c**) and β-tubulin displacement

vectors (**d**) shown from the tangential direction. **e** Structural changes of β-tubulin upon forming the one side of "MT-bond" with neighboring tubulin heterodimer. Two neighboring tubulin heterodimer models forming "MT-bond" in the Tube lattice are colored with medium purple and orange. The GMPCPP-tubulin heterodimer model in solution is colored in cornflower blue. The structure model of GMPCPP-tubulin heterodimer in solution is superimposed on the tubulin heterodimer (orange) in the Tube lattice, using α-tubulin as reference. The inset is a zoom-in view of key components of lateral contacts, including M-loop.

displacement of about 5 Å, further unbending the heterodimer into a fully straight conformation (Fig. 6e).

## Discussion

The dynamic instability of MT is a very fascinating phenomenon and very important process in all eukaryotic cells. Rooted in the center of the dynamic instability is the polymerization of tubulin induced by GTP, the hydrolysis of GTP into GDP upon polymerization, and the instability of GDP-tubulin inside the MT lattice[25,44]. How the nucleotide states affect the assembly properties and structures of tubulin remains enigma despite many studies trying to tackle the problem by different means. We now take advantage of the single-particle cryo-EM method to solve the tubulin heterodimer structure at its native state in solution without any co-factors that may influence the protein's conformation. These structures have revealed the sole effect of nucleotide bound to β-tubulin, therefore, directly verifying the allosteric hypothesis by GTP or GDP.

Our cryo-EM strcutures show that both GDP- and GMPCPP-tubulin heterodimer have a curvature of ~12° in solution, meaning that the nucleotide does not cause dramatic intra-dimer curvature changes (Fig. 1a–d). Comparing the cryo-EM structure of tubulin intra-dimer with those previously reported structures, we haven't observed any significant differences (Supplementary Fig. 4a, b). Based on our dataset, we have noticed an overall displacement of the β-tubulin relative to α-tubulin when performing structural comparison between different tubulin heterodimer structures (Supplementary Figs. 1c, e and 2d). It shows that GMPCPP-bound structures have less motion of β-tubulin (<1 Å) than GDP-bound structures (around 2 Å). In our view, these displacement values indicate the different intrinsic intra-dimer flexibility of tubulins. Compared to GMPCPP-tubulin, GDP-tubulin is more

flexible, leading to its larger displacemnent of β-tubulin. In addition, some of the most divergent regions, including the H2-S3 loop, are located close to the β-phosphate of GDP (Fig. 1d). Similarly, the poor continuity of the density map of M-loop and H2-S3 loop in GDP-bound state also reflects its larger structural flexibility (Fig. 1e–g). We speculate that the increase in flexibility is due to the loss of γ-phosphate. Further MD simulations confirms that the GTP lowers the RMSF value of its nucleotide binding domain (Supplementary Fig. 5). GTP binding, therefore, stabilizes the GTP pocket and the surrounding regions rather than leading to significant structural differences, thereby reducing structural flexibility and increasing homogeneity.

Consistently, the inter-dimer interface of GMPCPP state is also less divergent than that of GDP state. Even though both of GDP- and GMPCPP-tubulin tetramers swing in the radial bending direction, only the former exhibits tangential bending variation. (Supplementary Movies 1–4). The reduced flexibility of the inter-dimer interface may be a result of the reported flip out conformation of T5 loop[31,32]. The T5 loop extends out and makes contact with the tubulin heterodimer above, leading to more stable longitudinal interaction and less variation around the inter-dimer interface. When T5 loop does not flip out, GDP-tubulin tetramer shows a wider bending angle range in every direction, especially in the tangential direction, which has a major impact on the lateral interaction with adjacent dimers for both the Tube-bond and MT-bond interfaces. While it is possible that a greater degree of tangential bending flexibility may increase the chance of tubulin dimers coming into contact with each other, it does not guarantee the formation of preferred stable interactions. In fact, these contacts may dissociate shortly after collision, indicating a lack of stability. On the other hand, the relatively rigid conformation of GMPCPP-tubulin is more conducive to the establishment of stable

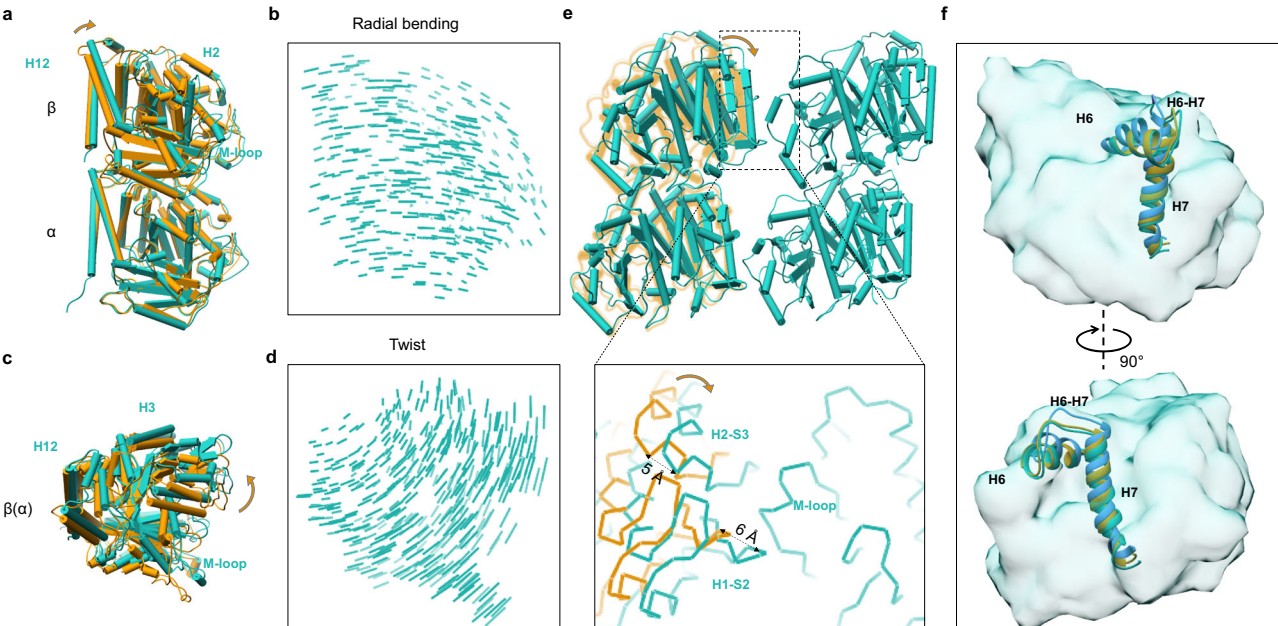

**Fig. 6 | H1-S2 and H2-S3 loops are pulled close to the neighbor M-loop to form the other "MT-bond". a, b** Structural comparison between the tubulin hetero-dimer in the Tube lattice and MT lattice, shown from the radial bending direction. Models are superimposed on the α-tubulin. The radial bending direction is indicated by the orange arrow. Two models are displayed in (**a**) as a side view, and the corresponding β-tubulin displacement vectors are shown in (**b**). **c, d** Structural comparison shown from the top view. Two models are displayed in (**c**), and the corresponding β-tubulin displacement vectors are shown in (**d**). **e** Structural changes of β-tubulin upon forming the other side of "MT-bond". Two adjacent tubulin heterodimer models in the MT lattice are colored in green. The structure model of tubulin heterodimer in the Tube lattice (orange) is superimposed on that of the left tubulin heterodimer in the MT lattice, using α-tubulin as reference. The inset is a zoom-in view of key components of lateral contacts of "MT-bond". **f** H6 and H7 helix of β-tubulin during different MT assembly stages. Top: Side view. Bottom: Back view. H6 and H7 helix of β-tubulin in solution is colored in cornflower blue, those in the Tube and MT lattice are colored in orange and green, respectively. The whole β-tubulin is shown with a green transparent surface.

lateral interactions. In addition, we have determined that the ratio of tubulin dimer, tetramer, and hexamer is not evidently altered between GDP and GMPCPP states (Supplementary Fig. 2c), thus ruling out the possibility that GDP-tubulin is not favoring MT assembly due to unfavorable formation of tetramer or hexamer itself.

While our previous work using mammalian tubulin allowed us to observe certain intermediate structures at higher magnesium concentrations (15–20 mM Mg$^{2+}$)[23–25,44], the use of *Drosophila* S2 tubulin enabled us to further explore the formation of helical ribbon and tube structures even at magnesium concentrations closer to physiological conditions. This difference can be attributed to the natural selection of insect tubulin, which possesses a higher polymerization property even at relatively lower temperatures (Supplementary Fig. 15). Therefore, by employing *Drosophila* S2 tubulin in our experiments, we are able to gain valuable insights into the assembly intermediates of microtubules under conditions that more closely resemble the physiological environment.

Our Cryo-EM structure of the S2-GTPγS-Tube demonstrates that the tubulin heterodimer adopts a curvature of ~6°, between the initial curvature of ~12° in solution (before assembly) and the final curvature of ~0° in MT (after assembly). There are two different lateral interactions co-existing in the Tube, named "Tube-bond" and "MT-bond". The MD simulation results show that the "Tube-bond" forms once two tubulin heterodimers encounter, which is triggered by the electrostatic interaction of R339 with five negative charged residues on the opposite surface of the neighbor tubulin (Fig. 4d and Supplementary Fig. 13a). The binding free energy of "Tube-bond" (−26.23 kcal/mol) is much weaker than that of "MT-bond" (−45.40 kcal/mol), indicating that the fast-formed interaction of "Tube-bond" is a sub-stable contact than the "MT-bond" (Supplementary Table 2). It soon begins to dissociate and converts into "MT-bond" (Fig. 4d–g and Supplementary Movie 6). However, the

process would probably take shorter than 3-4 μs to complete, therefore it is hard to observe the transition in natural conditions (Supplementary Fig. 14). Employing Go model-based MD simulations has afforded us this vivid depiction of the formation process of lateral interactions.

Both the Tube and MT can nucleate with 1.5 mM GTPγS and co-exist in 2 mM Mg$^{2+}$, but with more Tubes in conditions with higher concentration of Mg$^{2+}$ (Supplementary Fig. 9a, b), so the Mg$^{2+}$ concentration is the key factor to keep the intermediate structure. We speculate that the high concentration of Mg$^{2+}$ may stabilize the "Tube-bond" interface or prevent the transition of "Tube-bond" to "MT-bond". Indeed, MD simulations have revealed that Mg$^{2+}$ can bind to E330, stabilizing the "Tube-bond" interface, while E330 forms salt bridges with R214 at the "MT-bond" interface. (Fig. 4a, b and Supplementary Fig. 16). Consistently, when transferring the pre-formed Tube into the conditions with lower concentration of Mg$^{2+}$, we have observed the "Tube-to-MT" conversion. This suggests that a decrease in the Mg$^{2+}$ concentration triggers the conversion of "Tube-bond" to "MT-bond". As Mg$^{2+}$ concentration in a cell may vary, it would be worth examining MT assembly processes in vivo under different local or overall Mg$^{2+}$ concentrations to understand the regulation of MT dynamics by Mg$^{2+}$ flux.

The tubulin heterodimer goes through a large conformational change of "curve-to-straight" during the assembly process. Accompanying with the curvature decrease of tubulin heterodimer, β-tubulin's intermediate domain containing H6 and H7 helices has a rotation relative to the N- and C-terminal domains[27]. However, whether it is induced solely by the lateral contacts or the GTP binding, has been a controversial argument[21,23,56]. Recent simulation studies have further substantiated the notion that lateral contacts serve as crucial factors in the assembly of microtubules, exerting their influence through diverse mechanisms[42,43].

Here, the solution structures of tubulin heterodimer in different nucleotide states do not reveal a major conformational change between the GDP- and GMPCPP-tubulin except for the different flexibility. Instead, major heterodimeric curvature decrease happens in the Tube and MT assemblies (Fig. 6f). Therefore, the switch form curved to straight conformation is induced mostly by the lateral interaction. Further, we are interested in which lateral interaction contributes to the conformational change. The MD simulations shows that the curvature changes in nearly 2.5 μs, corresponding exactly to when the "MT-bond" forms, indicating a two-step straightening process of the tubulin heterodimer (Supplementary Fig. 14).

Comparing the tubulin heterodimer structure in solution and in the Tube lattice, we have found that one side of the "MT-bond" brings a half curvature change and a rotation of intermediate domain, where the M loop changes from unstructured state to an ordered helix (Fig. 5). During the comparison of tubulin heterodimer structure in the Tube and MT lattice, we have observed that the other side of the "MT-bond" formation is accompanied with the remaining curvature change and a movement of the N-terminal domain, including the H1-S2 and H2-S3 loops (Fig. 6). Therefore, the curvature changes are mostly induced by lateral interaction, rather than the bound nucleotide. But the more homogenous conformation of the heterodimer and tetramer induced by GTP clearly favors the fast formation of "Tube-bond" that later converts into more stable "MT-bond".

Additionally, we have also compared GTPγS-Tube with the previously reported GMPCPP-Tube structure. The MT-bond interface is similar, while the Tube-bond interface differs (Supplementary Fig. 17a–d). According to our structural and MD data, M-loop, H10-S9 loop, H9-S8 loop, H6 helix, H9 helix and H10 helix of one tubulin interact with the neighboring H3 helix, H4 helix, H5 helix and H4-S5 loop (Figs. 3f and 4a). According to the reported GMPCPP-tube structure, H10-S9 loop and H9-S8 loop of one tubulin interact with H3 helix, H4 helix and H4-S5 loop of the neighboring tubulin[23–25,44]. There is substantial overlap between these two Tube-bond interfaces. However, the Tube-bond interface of GTPγS-Tube has a larger buried area (468 Å$^2$) than that of GMPCPP-Tube (175 Å$^2$). One side of tubulin heterodimer in the GMPCPP-Tube interface has shifted about 20 Å related to that in the GTPγS-Tube bond interface (Supplementary Fig. 17d). We consider that different Tube-bonds may correspond to different intermediate states of lateral contacts. It is possible, for example, that the GMPCPP state mimics the very beginning state, followed by the GTPγS state. Due to the differences in lateral contacts, the tubulin dimer structure has minor structural differences (Supplementary Fig. 17e and f).

In raw cryo-EM micrographs and further data analysis, we have observed dimers, tetramers, and hexamers of tubulin, but no double- and triple-strand tubulin assemblies (Supplementary Figs. 1a and 2a). This means that the longitudinal interaction within a protofilament forms faster and more stable than the lateral interactions during early MT assembly stages. The critical length of protofilaments for a successful formation of lateral interaction should be at least greater than a hexamer. Consistently, the critical nucleus size of single-stranded oligomers was reported as 32 nm, corresponding to the length of a tubulin octamer[31]. The statistics for negative staining EM data showed that when a single-stand tubulin grows into octamer length, it begins to make lateral contacts with the neighbors[31]. Further investigations of the critical length of an octamer for initiating lateral contacts may be needed. Depending on its design, it may provide a larger contact surface or be slightly less flexible in tangential directions, etc.

Based on our results here, we have proposed that the lateral interaction in the early stages of MT assembly proceeds in several steps (Fig. 7). Three states are defined to describe the process of establishing lateral contacts. First, "the encounter state": the "Tube-bond" forms when a free tubulin dimer encounters a tubulin octamer, which is dynamically faster but not thermodynamically stable. The "Tube-bond" interface begins to dissociate before the intra-dimer curvature and its conformation start to change. Second, "the transient state": β-tubulin's intermediate domain has a downward rotation to α-tubulin so that the M-loop can be locked and stabilized into the pocket formed by the H1-S2 and H2-S3 loops in the N-terminal domain of neighboring tubulin. One side of "MT-bond" brings a half curvature change of the newly joined tubulin heterodimer. Third, "the stable state": the H1-S2 and H2-S3 loops move closely to the neighbor M-loop to form the other side of "MT-bond". The slightly curved tubulin becomes fully straightened with its both sides forming "MT-bond". In conclusion, GTP stabilizes tubulin heterodimer into more homogenous conformation and strengthens longitudinal interaction within a protofilament, both favoring the fast formation of a lateral "Tube-bond". The conversion of "Tube-bond" into thermodynamically more stable "MT-bond" further straightens tubulin dimer in the MT lattice.

In eukaryotes, microtubules are a major component of the cytoskeleton. Tubulin amino acid sequences are highly conserved across diverse eukaryotes such as budding yeast, protozoa, fruit flies, nematodes, unicellular algae, higher land plants, mice, and humans. Numerous studies have identified mutations in a wide variety of tubulin isotypes that lead to abnormal phenotypes[57]. As these residues are located throughout α- and β- tubulin, they may affect GTP binding pockets, the longitudinal interface, the lateral interface, the microtubule associated proteins (MAPs) binding sites and other regions. The detailed mechanism, however, remains unclear.

While we recognize the limitations arising from our utilization of MM/GBSA calculations and the constraints posed by low-resolution data, we appreciate that our research has cast a meaningful light within these confines. We concede that acquiring higher-resolution data in the future will undoubtedly enhance our understanding of the intricate underlying mechanism. It is noteworthy that, notwithstanding these limitations, our MD simulations has furnished significant insights, particularly in identifying potential key residues that play a role in the lateral interactions of both the "MT-bond" and "Tube-bond" states (Supplementary Table 2). We have found that a variety of phenotypes are associated with the mutation of these conserved residues in different isotypes. Additionally, these residues are highly conserved among different species (Supplementary Fig. 18). Based on the corresponding residues in our structures, mutations E113A, R123A, K156A, E160A, R215A, R284A, K299A, R311A and K338A in β-tubulin have resulted in recessive lethality and altered resistance to benomyl in *S. cerevisiae*;[58] mutation P289L in β-tubulin has produced abnormal MT arrays in *A. thaliana*[59]; α-tubulin mutation R308L has produced short seeds in *O. sativa*[60]; α-tubulin mutation R214H has caused cortical and cerebellar dysplasia in *H. sapiens*[9,10]; α-tubulin mutation R215C has also been found in individuals with familial amyotrophic lateral sclerosis (ALS)[61]. We anticipate further systematic and detailed research on this topic of how these residues function, particularly in "Tube-bond" interface. In addition, several cofactor proteins or small molecules may influence MT assembly at different stages of the process. The interfaces may provide potential targets to screen drugs to regulate MT dynamics in these tubulin states and their conversion.

## Methods

### Protein expression and purification
500 ml S2 cells (Invitrogen) were grown to $4 \times 10^6$ cells/ml in SF900II medium (Invitrogen), pelleted by centrifugation at $1700 \times g$ for 15 min, and resuspended in 1× BRB80 with 3 U of benzonase, 1 mM DTT, and protease inhibitors. After lysing the cells and centrifuging the lysate, the supernatant was applied to a pre-equilibrated TOG column containing 1× BRB80 and 100 μM Mg$^{2+}$ GTP. Elution was achieved using 3 column volumes (CV) of 1× BRB80, 10 μM Mg$^{2+}$ GTP, and 500 mM (NH$_4$)$_2$SO$_4$. Tubulin was subsequently desalted into 1× BRB80, 10 μM Mg$^{2+}$ GTP, concentrated to a minimum of 20 μM, and supplemented with 5% glycerol before being rapidly frozen in liquid nitrogen[62].

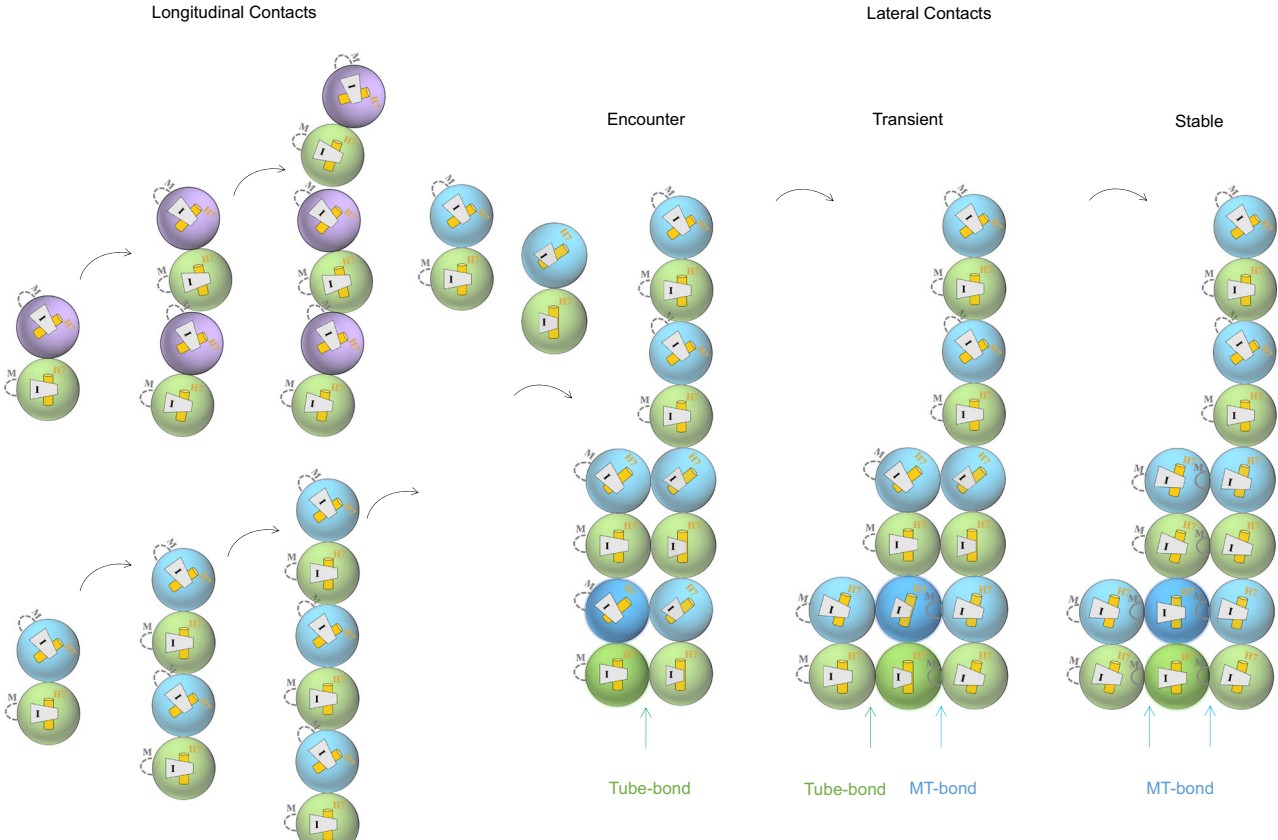

**Fig. 7 | Schematic illustration of the conformational changes of the MT assembly.** α- and β-tubulin are represented by green, blue (GTP bound) and purple (GDP bound) spheres. M-loops that are disordered and M-helixes that are ordered are indicated by curved dashed lines and solid lines, respectively. Helix H7 and the intermediate domain (I) undergo a rotational movement during the curved-to-straight process, indicating different intra-dimer curvatures. In the process of assembling microtubules, two major steps occur sequentially: longitudinal contacts are formed first, followed by lateral contacts. In the early assembly stage, single-strand GTP-tubulin oligomers display less structural variation between different tubulin heterodimers than GDP-tubulin oligomers. Upon reaching the length of four heterodimers longitudinally, a single strand of GTP-tubulin starts to recruit new tubulin heterodimer and form lateral contacts containing three different states: the "encounter state" (one Tube-bond), the "transient state" (one MT-bond), and the "stable state" (two MT-bonds). Once a newly joined tubulin heterodimer (colored with deep green and blue) has gone through the whole process mentioned above, the intra-dimer curvature changes from -12° to -0°.

Porcine tubulin was purified from porcine brain (Beijing No.5 Meat Processing Factory) through two cycles of polymerization and depolymerization[63], and stored at −80°C for usage after further purification using a TOG-based affinity column.

The construct of Rat Kinesin K560 (Kif5b, 1–560 a.a) was truncated to K349 (1–349 a.a) which only contains a single motor domain, and Glu236 was mutated to Ala (E236A) to eliminate its ATP hydrolysis but harbor the tubulin binding affinity[64]. The mutated Kinesin K349 (E236A) was purified as following[65]. Kinesin was purified from BL21 Escherichia coli strain (TIANGEN) by inducing expression at 22 °C. Cell lysis was performed in 50 mM phosphate buffer, pH 8.0, 300 mM NaCl, 2 mM MgCl₂, 10% (vol/vol) glycerol, 10 mM imidazole, and protease inhibitors. Clarified lysate was incubated with equilibrated nickel beads for 2 h. Beads were washed, and protein was eluted with 50 mM phosphate buffer, pH 8.0, 300 mM NaCl, 2 mM MgCl₂, 10% (vol/vol) glycerol and 250 mM imidazole. Concentrated kinesin was loaded onto an S200 Superdex column equilibrated in 25 mM Tris, pH 7.5, 150 mM KCl, 10% (vol/vol) glycerol, 2 mM MgCl₂, and 1 mM dithiothreitol (DTT).

**MT polymerization and Tube formation**

To assemble S2-GTPγS-MTs, S2-tubulin (4 mg/ml in stock) was reconstituted at 1.5–2 mg/ml in BRB80 buffer (80 mM 1,4-piperazinediethanesulfonic acid [PIPES], pH 6.9, 1 mM ethylene glycol tetraacetic acid [EGTA], 1 mM MgCl₂) supplemented with 1.5 mM GTPγS (Jena Bioscience, Germany), 1 mM DTT and 5% DMSO (vol/vol).

The mixture was then incubated at 28 °C for 2–4 h for S2-GTPγS-MTs to assemble.

GTPγS-Tube was obtained by incubating S2-tubulin (2-3 mg/ml) and GTPγS (1.5 mM) at 37 °C for 4–6 h, with a Mg²⁺ concentration of 5–10 mM. All GTPγS-Tube cryo-samples used for data collecting were prepared in buffer containing 5 mM Mg²⁺.

**Microtubule dynamic assay**

Tubulin was labeled with biotin (Thermo Fisher Scientific), TAMRA (Thermo Fisher Scientific) and Alexa Fluor 647 (Thermo Fisher Scientific) using NHS esters. GMPCPP microtubules (5% Alexa Fluor 647 labeled and 20% biotin labeled) were stabilized on the surface of the cover glass coated with a biotin-binding protein[62,66]. Porcine tubulin and *Drosophila* S2-tubulin were used to polymerize GMPCPP microtubules in the porcine and S2 microtubule dynamics assays, respectively. Dynamic microtubules started to grow from GMPCPP microtubules under 35 °C when porcine tubulin or *Drosophila* S2-tubulin was added to the flow cell. BRB80 supplemented with 2 mM GTP, 50 mM KCl, 0.15% sodium carboxymethylcellulose, 80 mM D-glucose, 0.4 mg/ml glucose oxidase, 0.2 mg/ml catalase, 0.8 mg/ml casein, 1% β-mercaptoethanol, 0.001% Tween 20 was used as the imaging buffer in our microtubule dynamic assay. The dynamics of microtubules was recorded by a total internal reflection (TIRF) microscope (Olympus) equipped with an Andor 897 Ultra EMCCD camera (Andor, Belfast, UK) using a 100× TIRF objective (NA 1.49;

Olympus). The growth rate and catastrophe frequency of dynamic microtubules can be measured using Fiji v1.53c[67].

## Negative stained sample preparation

2-3 mg/ml tubulin mixtures mentioned above were incubated in the BRB80 buffer supplemented with 10 mM $Mg^{2+}$ and 1.5 mM GTPγS at 37 °C for 5-6 hr to assemble GTPγS-Tube. The Tube was then collected by centrifugation (10,000 g, 10 min), and the pellet was washed three times mildly to remove unassembled tubulin, and resuspended in BRB80 buffer (only 1 mM $Mg^{2+}$ included). The resuspending sample continued to stay at 37 °C for another 2 h and we prepared negative stained samples at different time points. For negative staining, we added 3 μl the solution onto freshly glow-discharged EM grids coated by continuous carbon film, waited for 1 min, and stained the sample by 2% uranyl acetate before blotting the grid with filter paper and drying the specimen by air.

## Cryo-EM sample preparation of tubulin heterodimer, GTPγS-Tube and MT

For tubulin heterodimer, porcine tubulin (5 mg/ml) was incubated with 1.5 mM GMPCPP on ice for 20–30 min, and then centrifuged at 10,000 g for 10 min, followed by gel filtration (Superdex™ 200 Increase 3.2/300, GE Healthcare). The peak fraction from the gel filtration was used to make cryo-specimens of GMPCPP-tubulin. The preparation of GDP-tubulin was similar to that of GMPCPP-tubulin, except the incubation with GDP. Homemade graphene grids (300 mesh Quantifoil Au R1.2/1.3 grids coated with single-crystal graphene)[68] was firstly glow-discharged for 12 s at low level in Harrick Plasma. 4 μl tubulin solution (1–2 μM) was then pipetted onto the graphene grids, and blotted by filter papers (Ted Pella, Inc.) for 1 s at 8 °C with 100 % humidity in FEI Vitrobot Mark IV, and flash-frozen in liquid ethane cooled with liquid nitrogen.

For GTPγS-Tube, 3.5 μl Tube sample was applied to the glow-discharged grids (Quantifoil Cu R1.2/1.3, 400 mesh). After a 40 s incubation on the grid, 2 μl 3 mg/ml KMD in BRB80 buffer [80 mM Pipes (pH 6.9), 1 mM $MgCl_2$, 1 mM EGTA, 1 mM Dithiothreitol] was added. Then the mixtures were incubated on the grid for additional 40 s. To make full KMD decoration, another 2 μl KMD was added onto the grid and incubated for 40 s. The grid was subsequently blotted for 4 s using filter paper (Ted Pella Inc.) at 28 °C with 100% humidity, and plunged into liquid ethane with the equipment of Vitrobot Mark IV (Thermo Fisher Scientific).

The preparation of MT-KMD cryo-sample was mostly identical to that of Tube-KMD sample mentioned above.

## Cryo-EM data collection and processing

**Tubulin data.** Cryo-EM data were collected on a Thermo Fisher Titan Krios G3i electron microscope (300 kV) equipped with a Gatan K3 direct electron counting camera. Micrographs were recorded with a defocus range from −1.2 to −1.8 μm, in super-resolution mode at a nominal magnification of 81,000, corresponding to a final pixel size of 0.856 Å/pixel. The movie stacks containing 32 sub-frames were acquired using EPU with a dose rate of 1.56 e⁻/Å² per frame. 17,424 micrographs of GDP-tubulin sample and 14,503 micrographs of GMPCPP-tubulin sample were collected by EPU and then corrected by MotionCor2[69]. CTF estimation were performed using CTFFIND4[70].

For tubulin dimer data processing, all the following steps were performed in RELION3.1[71,72]. Particles were template-based autopicked and then imported for 2D classification. After 2D classification, particles belonging to good 2D averages were kept for further reconstruction. The initial model was derived from the reported crystal structure (PDBID: 4ffb) and then low-pass filtered to 20 Å[29]. After several rounds of 3D classification, GDP-tubulin yielded three good classes and GMPCPP-tubulin yielded two good classes. Then we did further 3D refinement respectively. According to the further structural

comparison results, GDP-tubulin has generated two different states, named GDP-1 and GDP-2. GMPCPP-tubulin has generated one final state. Finally, for GMPCPP-tubulin, we used 236,436 particles to get a reconstraion at 3.5 Å resolution. For GDP-tubulin, 287,272 particles were used to get a reconstruction of GDP-1 at 3.6 Å resolution and 143,422 particles were used to get a reconstruction of GDP-2 at 3.9 Å resolution, respectively.

For tubulin tetramer reconstruction, we used RELION3.1 and cryoSPARC v3.1.0[71–73]. Particles were picked via TOPAZ v0.2.4[74]. After several rounds of reference-free 2D- and 3D-classification (ab initio), 255,631 and 170,885 particles were kept for GDP-tubulin and GMPCPP-tubulin, respectively, which were imported into RELION3.1 for further 3D-classificaion. The initial model was calculated de novo. There were three major classes existing in both GMPCPP- and GDP-tubulin. We performed further local refinement for each class. For data processing of tubulin hexamer, the flowchart was similar to tubulin tetramer mentioned above. Particles were also picked via TOPAZ v0.2.4[74]. 149,126 and 133,737 particles were kept after 2D classification.

**Tube data.** Images were collected on a Titan Krios (300 kV) equipped with a Gatan K2-Summit direct electron-detecting device and GIF-quantum energy filter (Gatan) using AutoEMation[75]. Micrographs were recorded in super-resolution mode at a nominal magnification of 22,500, corresponding to a final pixel size of 1.33 Å/pixel. The total exposure time was 8 s, and each movie stack contained 32 sub-frames. The dose rate was 1.56 e⁻/Å² per frame and the accumulated dose in each stack was about 50 e⁻/Å². The defocus ranged from −0.5 to −2.5 μm. The flowchart of motion correction and CTF estimation was the same as the tubulin dataset processing.

Helical reconstruction was carried out with RELION3.0[76]. Filaments were picked manually, and segments were extracted using a box size of 800 pixels and an inter-box distance of 17.87 Å (the length of one asymmetric unit). Reference-free 2D classification was performed to remove bad particles. The initial helical symmetry parameter was calculated based on the 2D class average and diffraction pattern manually, following the previously reported protocol[77]. The helical parameters were then tested and refined in RELION and we got a final helical rise of 17.92 Å and a helical twist of 22.71° for the following 3D refinement. The first iteration of refinement was calculated using a cylinder as the initial model, and the resulted map was low-passed to 30 Å and employed as the initial model for a 3D classification with four classes. The most homogeneous particles were selected for further 3D auto-refinement. Finally, we solved the structure of GTPγS-Tube at a resolution of 6.8 Å using 74,919 particles.

**MT data.** Images were collected on a Titan Krios (300 kV) equipped with a Falcon II direct electron-detecting device. Micrographs were collected at a defocus range of −1.0 to −2.5 μm with the final pixel size of 1.08 Å/pixel. The total exposure time was 1.6 s, and each movie stack contained 26 sub-frames. The accumulated dose in each stack was about 50 e⁻/Å². The flowchart of motion correction and CTF estimation was the same as the tubulin dataset processing.

As the majority of S2-GTPγS-MTs were 15-pf which is a perfect helix without seam-line, we did data processing using helical reconstruction methods in RELION 3.0[76]. Particles were template-based autopicked and imported for reference-free 2D classification. Good particles were selected for 3D classification. Three references with different pfs (13pf, 14pf, 15pf) were given in the 3D classification. After 3D classification, the major class with 15pf MTs was sorted out for further refinement[78]. Finally, we used 25,448 particles to get the reconstruction at a resolution of 4.3 Å.

## Atomic model refinement and analysis

Porcine tubulin model was derived from the reported crystal structure (PDB ID: 6TIS and 6TIY) initially[31], and then adjusted manually in COOT

0.9[79] and finally refined in PHENIX 1.19.1[80]. The atomic model of S2 α- and β-tubulin was generated in SWISS-MODEL[81]. Then the α- and β-tubulin monomer were docked into the GTPγS-Tube and MT map by rigid body fitting in Chimera-1.13[82,83]. Then finally refined with real_space_refine in Phenix 1.19.1[80]. UCSF Chimera-1.13 were used for molecular graphics illustrations. Displacement vectors are generated and RMSD is calculated as described previously[49,51].

## Calculation of bending angle in three directions

Firstly, we use a tubulin tetramer model derived from the straight model solved by cryo-EM (PDB ID: 3JAR) as a reference. Then bending angle is calculated from the rotational and translational component of the transformation required to superimpose its β2-tubulin chain onto that of the reference model.

## Statistical significance analysis

Considering the non-normal distribution of both datasets and the limited number of values (only three) within each dataset, we expanded our datasets by creating 10000 groups of new datasets based on the existing samples by bootstrap. The objective was to conduct a hypothetical test while ensuring the newly generated datasets maintained the variance as the original datasets but with a larger number of samples. This expansion aimed to provide a more comprehensive dataset, facilitating a more robust statistical analysis.

To assess the statistical significance of std of these datasets, we performed a two-tailed *T*-test. This test was selected due to their suitability for normal distribution of the std of generated datasets. The *T*-test and corresponding *p*-values for each case, were calculated using the Scipy toolbox[84] in Python3[85].

## 3D variability analysis

3D variability analysis was exploited in cryoSPARC v3.1.0[73]. We firstly performed homogeneous refinement using 244,000 and 168,205 particles of GDP- and GMPCPP-tubulin respectively, followed by 3D variability analysis (3DVA) with a filter resolution of 8 Å. The 3DVA display job was set with the output mode of 20 intermediate frames[86].

## Quantification of tubulin dimer, tetramer and hexamer particles

We traced back raw micrographs containing hexamer particles based on their 2D classification results. In cryoSPARC v3.1.0[73], we auto-picked particles with tubulin dimer, tetramer, and hexamer particles with templates, ran an iteration of 2D classification, and then removed those classes derived from noise and ice contamination. The GDP state dataset contains 4,252 tubulin heterodimer particles, 2,222 tubulin tetramer particles, and 379 tubulin hexamer particles. As a percentage of the total particles per micrograph, tubulin dimers, tetramers, and hexamers account for 62%, 32%, and 6%, respectively. According to the GMPCPP dataset, there are 4,498 tubulin heterodimers, 1,990 tubulin tetramers, and 472 tubulin hexamers. The proportion of tubulin dimers, tetramers, and hexamers in each micrograph is 64%, 29%, and 7%.

## Molecular dynamic simulations

The initial coordinates of one αβ-tubulin with GTP in the intermediate conformation were built based on the models obtained from the cryo-EM structures of GTPγS-Tube. The missing residues of dimer were constructed using Modeller9.20[87], and three different initial conformations were selected for molecular dynamic simulations. The conformations of one dimer (GDP or GTP) in solution and two dimers (GTP) with an interface of Tube-bond or MT-bond were used in the simulations.

Atomistic molecular dynamic simulations of initial models were carried out in the AMBER18 program using AMBER14SB force field for protein[88,89] and the parameters for the nucleotides (GTP and GDP) were obtained from the parameters reported previously[90]. Each system was neutralized with a number of magnesium ions and then immersed in a solvent box filled with TIP3P water molecules[91], to warrant a distance of at least 10 Å between the surface of each protein models and the water box edge. The entire systems were subject to the energy minimization in three stages to remove the bad contacts. Firstly, the solvent and the neutralized ions were minimized by holding the protein and ligand using a restraint with strength of 100 kcal/(mol Å$^2$), and then the minimization was performed by holding the protein and ligand using a constraint of 10 kcal/(mol Å$^2$). Finally, the whole systems were minimized by removing any constraint. Each stage was performed using the steepest descent minimization of 1000 steps followed by a 9000 steps conjugate gradient minimization. NVT (constant Number of atoms, Volume and Temperature) simulations were carried out by heating the whole system linearly with time gradually from 100 to 300 K in the first 300 ps, and the Berendsen thermostat[92] was used to maintain the temperature of the whole system. Subsequently, the system was equilibrated under the temperature of 300 K for 1 ns was followed by a NPT (constant Number of atoms, Pressure, and Temperature) production run. During the heating stage, all the protein and ligands were restrained by a restrained of 100 kcal/(mol Å$^2$), and under equilibration stage the restraint strength was decreased to 10 kcal/(mol Å$^2$). During the NPT production run, the Berendsen barostat[93] was used to control the pressure at 1 atm, and the Langevin thermostat was employed to control the temperature of thesystems at 300 K. All bonds associated with hydrogen atoms were constrained by employing the SHAKE algorithm[94], and the Hydrogen Mass Repartitioning (HMR) method was adopted, such that the integration time step of 4 fs could be used. A cutoff value of 12 Å was set for nonbonded interactions and the Particle Mesh Ewald method[95] was employed for treating electrostatic interactions. For each system, five independent molecular dynamic simulations were carried out using different velocities that were randomly generated at the beginning of the simulations and run for 1 μs. The analysis of each molecular dynamic trajectory was performed with the cpptraj module in Amber 18[96].

The root-mean-square-fluctuation (RMSF) of a structure was calculated according to the following equation, where $X_i$ is the coordinates of particle $i$, and $\langle X_i \rangle$ is the ensemble average position of $i$.

$$\rho_i = \sqrt{\left\langle \left( X_i - \langle X_i \rangle \right)^2 \right\rangle} \tag{1}$$

Thus, the value of RSMF can reveal the flexibility of the simulating system. An area of the structure with high RMSF values frequently diverges from the ensemble average, indicating high flexibility. For the tubulin heterodimer model in GDP-1, GDP-2 and GTP states, the RMSF values were calculated using alpha-carbon atoms of all proteins with reference to the average conformation of MD simulations.

## MM-GBSA calculation

To understand the interaction between the two dimers, the binding free energies were calculated using the MM-GBSA method[97]. For each complex, 500 snapshots were extracted from the last 100 ns along the molecular dynamic trajectory at an interval of 200 ps. The binding free energy ($\Delta G$) can be represented as:

$$\Delta G = \Delta E_{MM} + \Delta G_{sol} \tag{2}$$

where $\Delta E_{MM}$ is the difference of molecular mechanic energy between the complex and each binding partner in the gas phase, $\Delta G_{sol}$ is the solvation free energy contribution to binding and $T\Delta S$. $\Delta E_{MM}$ is further divided into two parts:

$$\Delta E_{MM} = \Delta E_{ele} + \Delta E_{vdW} \tag{3}$$

where $\Delta E_{ele}$ and $\Delta E_{vdW}$ are described as the electrostatic interaction and van der Waals energy in the gas phase, respectively. The solvation free energy is expressed as:

$$\Delta G_{sol} = \Delta G_{gb} + \Delta G_{np} \qquad (4)$$

where $\Delta G_{gb}$ and $\Delta G_{np}$ are the polar and non-polar contributions to the solvation free energy, respectively.

## Molecular simulation with all-atom structure-based model

To characterize the spontaneous assembly process of tubulin, additional simulations with an all-atom structure-based force field were performed. In these simulations, the potential energy of the system was defined to have a global minimum corresponding to the MT configuration, and the initial structure was built with three parallel tubulin dimers separated by ~15 Å. To remove the potential bias caused by the placement, three initial conformations with different directions were set.

The all-atom structure-based SMOG force field[98] could explicitly represent every non-hydrogen atom. It took advantage of harmonic potentials to describe the backbone geometry (bond lengths, angles and improper/planar dihedral angles) and cosine functions to represent flexible dihedral angles. Non-bonded contacts in the endpoint configuration were assigned 6–12 interactions to guide the assembly, while atom pairs that were not identified as contacts were assigned an excluded volume interaction. The force field files were generated using the SMOG2 software package[99], and the non-bonded contacts were identified through the Shadow Contact Map algorithm[100] with a cutoff distance of 6 Å.

Molecular dynamics simulations were carried out by OpenMM with OpenSMOG[101] plugin. In all simulations, the system was maintained at a temperature of 0.5 reduced units via Langevin dynamics protocols, and the timestep was set to 0.001 reduced units. For each initial structure, about 70 independent simulations ($2*10^7$ steps) were performed to ensure enough sampling of the assembly process. According to the comparison of diffusion coefficients in SMOG force field and all-atom explicit-solvent force field, the effective simulated time of each simulation could be ~10 μs[102].

## Reporting summary

Further information on research design is available in the Nature Portfolio Reporting Summary linked to this article.

## Data availability

Atomic coordinates and their corresponding maps were deposited in RCSB under the accession code: GMPCPP-tubulin in solution (EMD-34077, PDB: 7YSN), GDP-1 tubulin heterodimer in solution (EMD-34078, PDB: 7YSO), GDP-2 tubulin in solution (EMD-34079, PDB: 7YSP), GTPgammaS-Tube-KMD (EMD-34080, PDB: 7YSQ) and GTPgammaS-MT-KMD (EMD-34081, PDB: 7YSR). All other data and materials are available from the corresponding authors upon reasonable request. Previously solved structures used in this study were obtained from the PDB with accession codes: 3JAR; 3JAK; 6TIS; 6TIY; 4FFB; 4DRX. The source data for Fig. 2i,j,k, Supplementary Fig. 2c, 5a, 14a, 15b and 15c are provided in source file. Source data are provided with this paper.

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

## Acknowledgements

We thank K. Xu for his assistance in calculating the bending angles of tubulin tetramer. We thank Y.W. Li for advice on helical symmetry determination. We thank R. Zhang for kindly providing the scripts to calculate RMSD and displacement vectors. We thank Prof. L. Yu and W.Q. Du for kindly providing the original kinesin construct. We thank L.L. Wang for his help in KMD plasmid construction. We thank L.Y. Zhao for her help in MT data processing. We thank Z.Q. Guo and F.H. Meng (Shuimu BioSciences) for Cryo-EM data collection of tubulin dimer. We thank J.L. Lei, X.M. Li, F. Yang, X.F. Hu and J. Wen for cryo-EM data collection of GTPγS-Tube and MT in the National Protein Science Facility (Beijing) at Tsinghua University. We thank T. Yang, Y.K. Wang, A.B. Jia and J.H. Chen for providing the high-performance computation platform. This work has been supported by the National Natural Science Foundation of China (31825009 to H.-W.W.; 21933010 to G.L.), the Strategic Priority Research Program of Chinese Academy of Sciences (XDB 37000000 to G.L.)

## Author contributions

H.-W.W. conceived the project. J.Z. performed sample preparation, EM data processing, model refinement and structural analysis. H.-W.W., G.L. and H.C. designed the MD strategy and procedure. H.C., and A.W. performed the simulations. G.L., H.C. and A.W. performed the data analysis of the simulations. Y.S. and X.L. purified the tubulin and performed TIRF experiments. N.L. provided the graphene grid and assisted with the cryo-EM sample preparation of tubulin heterodimer. J.W. assisted with the data collection and data processing. Y.L. performed statistical significance analysis. J.Z., H.-W.W, H.C. and G. L. wrote the manuscript. All authors contributed to the final manuscript.

## Competing interests

The authors declare no competing interests.
