## [Peer Review File · Nature Communications]

REVIEWER COMMENTS

Reviewer #2 (Remarks to the Author):

This manuscript reports a comprehensive structural study that uses both single particle cryo-EM and molecular dynamics simulation to develop a fundamental understanding of the detailed process of microtubule assembly. The authors determined high resolution structures of porcine tubulin GDP heterodimers and GMPCPP heterodimers to enable pseudo-atomic model buildings. They have also obtained the 3D structures of the tubulin tetramers and hexamers of GDP and GMPCPP in solution. The structural results demonstrated that the γ -phosphate of GTP is positioned to stabilize the nucleotide binding domain, which results in a more stable and homogeneous tubulin dimer structure; such a structure leads to more rigid GTP tetramers comparing to more flexible GDP tetramers. The MD simulation results agree with this conclusion. These results indicate that the microtubule assembly with GTP-tubulin heterodimers is not due to a change in structural conformation associated with nucleotide status and thereby the straightness of the tubulin protofilaments. Rather, the GTP in β -tubulin helps to stabilize the structure of tubulin dimers and their oligomers for a more stiff and longer protofilament that favors the microtubule polymerization. It has been a widely accepted hypothesis/belief that the different nucleotide status between GTP- and GDP tubulin α - β -dimers may introduce differences in structural conformation and thereby changes in the curvature of tubulin protofilament, which affects the microtubule polymerization or depolymerization process. This manuscript provides structural evidence different from this understanding. This work reminded me of the recently published cryo-EM studies of actin filaments (Reynolds et al Nature 2022, Oosterheert et al. Nature 2022), which demonstrated that the nucleotide status of ATP/ADP-Pi/ADP have minimum influence on actin filament structures. It is very interesting that in both dynamic cytoskeleton filaments, microtubule and F-actin filaments, change in nucleotide status affects the polymerization process through a mechanism other than a structural conformational change. This result in the manuscript represents a significant advance in understanding of the polymerization process of microtubules.

In addition, the authors studied lateral interactions of tubulin dimers for understanding the microtubule assembly process. They successfully expressed and purified *Drosophila* S2 tubulins and studied the structure of a large S2-GTP γ S tube in addition to the microtubules of S2 GTP γ S tubulin. While the tubulin intra-dimer bending angles for tubulin dimers in solution, S2- GTP γ S tube, and S2-GTP γ S microtubule are 12°, 6°, and 0° respectively, the tubulin lateral interaction in the S2-GTP γ S tube (the "Tube-bond" in the manuscript) is proposed to be an intermedium lateral interaction before the final formation of the stable "MT-bond" for microtubule formation. Thus, the authors proposed a novel and plausible model for the microtubule assembly process.

Overall, this manuscript represents an excellent piece of science work, with a significant amount of research effort included, insightful new information provided, and a plausible microtubule assembly model proposed. The paper itself is well-organized and clearly written. I recommend it for publication

with minimum modification. There are several minor issues that I would like to bring to the authors' attention:

(1) Page 3, 2nd paragraph: "The GTP bound to α -tubulin is buried in the α -, β -tubulin intra-dimer interface and is non-exchangeable and non-hydrolysable, whereas the nucleotide bound to β -tubulin is exposed to solvent and exchangeable". Suggestion: "The GDP bound to α -tubulin is buried in the α -, β -tubulin intra-dimer interface and is non-exchangeable, whereas the GTP bound to β -tubulin is hydrolysable and exposed to solvent, and therefore exchangeable."

(2) To some readers, the cartoon picture in Fig.7 may not be very helpful for understanding the author's model of microtubule assembly process. The N, I, C three domain pattern of a tubulin monomer is unnecessary and may result in confusion to some readers. Using simple patterns would be a better option. It is very difficult to tell the differences in Fig. 7 between the three kinds of tubulin dimers, the free dimers in solution, Tube-bond and the MT-bond. For illustration purpose, the authors may consider using more exaggerated angles in the drawings to represent the 12-, 6-, and 0-degree bending other than using the accurate angles. If the authors take this suggestion and redesign and redraw Fig. 7, I suggest looking at the drawing of Box 2 and/or Figure 1 of the review paper by Anna Akhmanova and Michel O. Steinmetz (Nature Reviews Molecular Cell Biology, 2015) as a good example.

(3) The design of Movie S5 is very good for demonstrating the components packing purpose. However, some parts of this long movie can be more polished. For example, at the 59", the movie may use a smoother transition so that readers understand that the component is placed back in the whole tube. There are several places in the movie that share a similar issue and can be polished.

(4) There are previously published microtubule structural studies with some information about the intra-dimer interactions and/or inter-protofilament lateral interactions that mentioned salt bridge(s) like that reported in this manuscript. They seem to be neglected by this manuscript.

Reviewer #3 (Remarks to the Author):

This manuscript by Zhou et al. aims to uncover the mechanism behind the role of GTP binding and hydrolysis in microtubule (MT) assembly, a critical process in many eukaryotic cellular events. Through a series of cryo-electron microscopy structures and molecular dynamics simulations, the authors found differences in the flexibility of GDP- and GTP-bound tubulin heterodimers, and described sequential steps in the formation of lateral contacts during the early stages of MT assembly. They propose a flexible model of GTP-initiated MT assembly to explain the nucleation and assembly of MT. The findings are certainly important, as they offer new structural insight into the underlying mechanism of GTP binding and hydrolysis triggering MT assembly.

A major strength of the study is the use of single-particle cryo-electron microscopy (cryo-EM) to provide detailed tubulin heterodimer and oligomer structures as a function of nucleotide state. To the best of

my knowledge, this is the first study that has resolved the in-solution structure of a single tubulin dimer without the addition of MT-depolymerizing drugs or co-factors (such as stathmin). The manuscript also markedly advances the field by describing the 3D structure of short tubulin oligomers, which offers a more complete picture of the assembly process than previous 2D image analyses (e.g., Ayukawa et al. 2022). These oligomeric tubulin structures will also enable quite informative comparisons to future tubulin oligomers simulations. This is certainly a major breakthrough and should definitely be published.

However, I have some major issues, particularly regarding the second part of the manuscript, which need to be addressed before publication.

(1) Extraction of flexibility from the observed three cryo-EM classes. The authors draw conclusions about the difference between the flexibility of GDP- and GTP-(analog)-tubulin tetramers from the ‘variation range’ observed in three cryo-EM classes each, without considering the different observed populations and without considering how structural classes that may not have been observed (e.g., in ED Fig 5 there is an additional 5% class that is disregarded) might affect the result. Also, it is unclear to what extent the used rigid body fitting of dimers might affect the result in comparison to flexible fitting (e.g., Rosetta, CDMD, MDFF), which this referee would think is more appropriate here. Also, the fits are ensemble averages and it is unclear whether the average over the full ensemble (as opposed to 3 class averaged structures) would yield similar results.

Given these uncertainties, it is hard to see why the difference between a range $[-44^\circ, -20^\circ]$ and $[-41^\circ, -24^\circ]$ should point to a significant difference in flexibility, particularly because much of the ranges are due to the least populated classes. Perhaps, as a better alternative, the authors might consider reporting the weighted standard deviation of the respective angles, which, for the radial bending, yields $\sim 7.7^\circ$ for GDP and $\sim 6.4^\circ$ for GMPCPP. To establish that this difference is indeed significant, the authors will need to provide a conservative error estimate for these angles. Also, this referee was unable to see why the combined angles lie outside the intervals defined by the respective three classes in most cases.

(2) Interpretation of different flexibilities. The mechanism derived from these observations is probably not the only possible one. For instance, if GDP-oligomers are more flexible tangentially, shouldn't it be easier to interact with neighboring oligomers laterally? Accordingly, the combined data might as well suggest a different mechanism: because GDP-oligomers have, on average, a much larger tangential deformation than GMPCPP-oligomers, they are less likely to form a proto-lattice where all dimers could adopt a regular “MT-bond”.

(3) Why do the authors provide no characterization of the observed tubulin hexamers? The hexamers encompass two inter-dimer bonds and their conformation would in theory be less affected by short-size artifacts. This is a valuable piece of information and could also serve to estimate the required uncertainties above.

(4) Section “The conversion from Tube-bond to MT-bond.” The MD simulation part is based on the assumption that tubulins assembling into proto-lattice oligomers indeed undergo the observed conformational transition, which should be stated. Also, the authors should better explain exactly which question the simulations try to address.

The authors claim that “the MD simulations demonstrate the initial formation of the Tube-bond followed by the conversion in the MT-bond.” In the supplementary material, they however write that “the potential energy” for their target structure (three laterally coupled tubulin dimers) “was defined to have a global minimum corresponding to the MT configuration.” It is not surprising that the authors observed the formation of, first, a Tube-bond between the three dimers followed by the formation of a MT-bond – indeed, the simulation was biased that way. So which additional information do these simulations provide in light of the main conclusions of this study? In my view, unless I overlooked something, the target-biased simulations could be omitted.

The authors also used the MM-GBSA method to estimate the free energies of Tube- and MT-bonds. This is a quite inaccurate method, which should be clearly stated. In this context it would be good to compare/validate the obtained free energies results against values published previously using different methods?

Minor issues and readability for general audience:

(5) Section “Capture of MT assembly intermediates from *Drosophila* S2 tubulin”.

The transition from bovine and porcine tubulin dimers and short oligomers to resolving the structure of “GTPgammaS-Tubes” made of *Drosophila* tubulin is probably difficult to understand for the non-expert. In particular, the following questions may arise:

* Why the switch of the tubulin species? If Tube and ribbon structures can't be observed with mammalian tubulin under the same conditions as for the oligomers, does this sufficiently justify the switch to insect tubulin?

* If we do observe them for *Drosophila* tubulin and GTPgammaS, can we really conclude that these are indeed structural intermediates of MT assembly and not alternative and/or misassembled forms of *Drosophila* tubulin?

* I understand that it is very hard to capture larger oligomers like laterally coupled hexamers, but isn't the step from a small oligomer to a tubulin-kinesin double-layer tube too large to draw conclusions about the initial steps of tubulin oligomerization?

(6) It would be useful to discuss how the results agree with recent simulations (e.g., Alexandrova et al, 2022. PNAS 119 (46) e2208294119, Igaev & Grubmüller, 2022. PNAS 119 (12) e21155161)?

(7) The methods part describing the MD-simulations lacks detail to an extent that the reported simulations are not reproducible. Examples are:

“Energy minimization was performed by imposing a strong restraint on each system” – how strong precisely?

“[...] by minimizing the whole system for a few thousand steps.” – how many steps precisely? Which method? Conjugate gradient? Steepest descent?

“[...] by heating the whole system slowly from 100 to 300 K,” – How long? Linearly with time or exponentially by a weak T-coupling constant?

Which pressure coupling constant was used? Which pressure? (I guess 1 Atm, but should be specified.)

(8) Lines 92-106: The description of previous work needs improvement. For example, not all of the cited studies used MD simulations, contrary what is claimed in the text. The authors cite a very heterogeneous and somewhat unmotivated mix of papers that used phenomenological models of MT assembly, empirical mechanochemical models, coarse-grained MD, and unbiased all-atom MD as well as free energy calculations without clearly explaining the context. Also, the list of the cited simulation studies is missing some key contributions from the recent years.

Reviewer #4 (Remarks to the Author):

Zhou et al's manuscript addresses one of the fundamental questions in the cytoskeletal field, namely the mechanism of assembly of the microtubule polymer from soluble tubulin and how it is regulated by hydrolysis of bound GTP by the β -tubulin subunit. Much work has already been done on this problem, but, here, the authors take several new approaches that substantially contribute to the field. The authors report four distinct types of structure using cryo-electron microscopy, including (for the first time) structures of porcine tubulin dimers in a solution state bound to either GDP or the slowly-hydrolyzing GTP analog, GMPCPP, as well as spontaneously forming small oligomers of tubulin dimers (tetramers and hexamers). This represents a true technical feat, due to the small size of these structures and low frequency of the oligomers. In addition, they report the first structures of *Drosophila melanogaster* tubulin assembled into microtubules, as well as an analog of a late-stage MT assembly intermediate. Combining this panoply of structures with modeling using molecular dynamics, they provide a novel perspective on how MTs assemble. This work will be of great interest to the readers of Nature Communications, and we recommend its acceptance.

Our main critical comment is that, to make the work more accessible to a general audience, the results should be more explicitly situated into the prior, extensive literature on microtubule structure and microtubule assembly. Below, we have indicated some key places where additional figures or discussion will strengthen the article:

1. The authors state that determining the structure of soluble GDP and GMPCPP-bound tubulin will yield a more accurate picture of the MT building block than the previously determined crystal structures, which may be artificially constrained by crystal contacts and the addition of polymerization-inhibiting drugs. Now that the authors have solved the structure of soluble tubulin, a short discussion should be included of what differences, if any, are observed between these new structures and the prior crystallographic structures (similar to the discussion for the tubulin tetramers in lines 171-173). In addition, a superposition of the prior crystallographic tubulin tetramer structure(s) onto the ensemble of new tubulin tetramer structures either in Figure 2, or in a supplement, would aid the reader substantially.
2. The presence of tubulin tetramers and tubulin hexamers in both GDP and GMPCPP samples is very interesting and the comparison between the two structural ensembles is well done. It would be beneficial to the reader to understand how the frequency of tubulin tetramer and tubulin hexamer compares between the GDP and GMPCPP samples. Can the authors quantify the average number of tetramer particles per micrograph as a percentage of total particles or as a ratio with tubulin dimers? This is valuable information that would speak to the authors' model and could help substantiate the claim in the discussion (lines 404-405) that GDP-tubulin is unfavorable to initiate MT assembly due to a larger range of motion in the tetramer, and not because formation of tetramers themselves is

disfavored. The number of hexamer particles should also be quantified in the same way to provide a further data point. This information could be easily incorporated into Extended Data figures 1 and 2.

3. The use of *Drosophila melanogaster* tubulin is a clever strategy to circumvent temperature and kinetic limitations inherent to the standard mammalian tubulins. However, it raises the issue of whether results using *D. melanogaster* tubulin can be extrapolated to mammalian tubulin. At minimum, the authors should include the sequence identity between bovine and *D. melanogaster* tubulin in the main text (most likely in the Results subsection 'Capture of MT assembly intermediates from *Drosophila* S2 tubulin'), and also a sequence alignment, including all species mentioned in the Discussion paragraph 'Tubulin mutations and their associated abnormal phenotypes' and highlighting the amino acids found within the 'Tube-bond' and 'MT-bond.' This paragraph in the discussion should also either reference an existing figure or be accompanied by a new figure that highlights the residues mentioned.

4. The use of the kinesin-1 motor domain to improve and interpret the structure of the GTP γ S *D. melanogaster* tube is likewise very clever. However, given prior reports that addition of microtubule-binding proteins to *D. melanogaster* tubulin can alter assembly reactions, we were concerned that the kinesin motor domain might be influencing the resulting EM reconstruction, and thus that the kinesin-bound structure might not represent a true on-pathway assembly intermediate. Could the authors provide additional data that would clarify this point? The easiest option might be an overlay of the lower-resolution undecorated GTP γ S-Tube with the decorated higher resolution tube to show that the two structures are analogous. In addition, it would be beneficial to include a comparison between the structure of the previously published GMPCPP bovine tube (citation 46) within a figure. Note that this reference should be cited also at line 282.

5. The distribution of protofilament number among microtubules for the *D. melanogaster* GTP γ S-kinesin-1 structure is different from mammalian GTP γ S microtubules, in that it is dominated by 15-protofilament microtubules. Mammalian GTP γ S microtubules, however, are predominantly 14-protofilament (Zhang R, LeFrance B, Nogales E. PNAS 2018). The authors should comment on this, as it may be of interest to the readers.

Minor comments:

- The comparison of bovine and *Drosophila* tubulin dynamics (Extended Data Figure 13) is not referenced or discussed in the text. This could be added, eg, to the Results section around line 224.

- The temperature the dynamic assays were performed at, as well as other key assay conditions including Mg $^{2+}$ concentration, glycerol, buffer, etc, should be included in the Methods section and, potentially, in the legend of Extended Data Figure 13 (eg bovine tubulin polymerizes at a different optimal temperature than *D. melanogaster* tubulin, which does not make bovine tubulin intrinsically less capable of polymerizing). Also, it should be clarified whether the seeds the soluble tubulin is polymerizing from are bovine or *Drosophila*.

- Dithiothreitol (DTT) should be spelled out (line 878) the first time it is mentioned
- The versions of Fiji (line 890), Topaz (line 952), COOT (line 1001), PHENIX (line 1002), and Chimera (line 1004) should be specified
- In the discussion section 'A hypothetical model of MT assembly', the authors claim at line 476 that 'The statistics for negative staining EM data shows that...'. Are these statistics coming from citation 28? If so, that should be clarified.
- The paper will need to be generally proof-read for minor typos and grammar issues prior to publication

Responses to Reviewers

Response to Reviewer #2

This manuscript reports a comprehensive structural study that uses both single particle cryo-EM and molecular dynamics simulation to develop a fundamental understanding of the detailed process of microtubule assembly. The authors determined high resolution structures of porcine tubulin GDP heterodimers and GMPCPP heterodimers to enable pseudo-atomic model buildings. They have also obtained the 3D structures of the tubulin tetramers and hexamers of GDP and GMPCPP in solution. The structural results demonstrated that the γ -phosphate of GTP is positioned to stabilize the nucleotide binding domain, which results in a more stable and homogeneous tubulin dimer structure; such a structure leads to more rigid GTP tetramers comparing to more flexible GDP tetramers. The MD simulation results agree with this conclusion. These results indicate that the microtubule assembly with GTP-tubulin heterodimers is not due to a change in structural conformation associated with nucleotide status and thereby the straightness of the tubulin protofilaments. Rather, the GTP in β -tubulin helps to stabilize the structure of tubulin dimers and their oligomers for a more stiff and longer protofilament that favors the microtubule polymerization. It has been a widely accepted hypothesis/belief that the different nucleotide status between GTP- and GDP tubulin α - β -dimers may introduce differences in structural conformation and thereby changes in the curvature of tubulin protofilament, which affects the microtubule polymerization or depolymerization process. This manuscript provides structural evidence different from this understanding. This work reminded me of the recently published cryo-EM studies of actin filaments (Reynolds et al Nature 2022, Oosterheert et al. Nature 2022), which demonstrated that the nucleotide status of ATP/ADP-Pi/ADP have minimum influence on actin filament structures. It is very interesting that in both dynamic cytoskeleton filaments, microtubule and F-actin filaments, change in nucleotide status affects the polymerization process through a mechanism other than a structural conformational change. This result in the manuscript represents a significant advance in understanding of the polymerization process of microtubules.

In addition, the authors studied lateral interactions of tubulin dimers for understanding the microtubule assembly process. They successfully expressed and purified *Drosophila* S2 tubulins and studied the structure of a large S2-GTP γ S tube in addition to the microtubules of S2 GTP γ S tubulin. While the tubulin intra-dimer bending angles for tubulin dimers in solution, S2- GTP γ S tube, and S2- GTP γ S microtubule are 12°, 6°, and 0° respectively, the tubulin lateral interaction in the S2-GTP γ S tube (the “Tube-bond” in the manuscript) is proposed to be an intermedium lateral interaction before the final formation of the stable “MT-bond” for

microtubule formation. Thus, the authors proposed a novel and plausible model for the microtubule assembly process.

Overall, this manuscript represents an excellent piece of science work, with a significant amount of research effort included, insightful new information provided, and a plausible microtubule assembly model proposed. The paper itself is well-organized and clearly written. I recommend it for publication with minimum modification. There are several minor issues that I would like to bring to the authors' attention:

We thank the Reviewer for being enthusiastic about our study and for recognizing its novelty and significance. We have revised the manuscript to address all the points raised in the review, as described below.

Q1. Page 3, 2nd paragraph: "The GTP bound to α -tubulin is buried in the α -, β -tubulin intra-dimer interface and is non-exchangeable and non-hydrolysable, whereas the nucleotide bound to β -tubulin is exposed to solvent and exchangeable". Suggestion: "The GDP bound to α -tubulin is buried in the α -, β -tubulin intra-dimer interface and is non-exchangeable, whereas the GTP bound to β -tubulin is hydrolysable and exposed to solvent, and therefore exchangeable."

R1. We have revised our manuscript to better illustrate GTP and GDP nucleotides with non-exchangeable and exchangeable sites. "The GTP bound to α -tubulin is buried in the α -, β -tubulin intra-dimer interface and is non-exchangeable and non-hydrolysable, whereas the nucleotide bound to β -tubulin is exposed to solvent and exchangeable."

Q2. To some readers, the cartoon picture in Fig.7 may not be very helpful for understanding the author's model of microtubule assembly process. The N, I, C three domain pattern of a tubulin monomer is unnecessary and may result in confusion to some readers. Using simple patterns would be a better option. It is very difficult to tell the differences in Fig. 7 between the three kinds of tubulin dimers, the free dimers in solution, Tube-bond and the MT-bond. For illustration purpose, the authors may consider using more exaggerated angles in the drawings to represent the 12-, 6-, and 0-degree bending other than using the accurate angles. If the authors take this suggestion and redesign and redraw Fig. 7, I suggest looking at the drawing of Box 2 and/or Figure 1 of the review paper by Anna Akhmanova and Michel O. Steinmetz (Nature Reviews Molecular Cell Biology, 2015) as a good example.

R2. We appreciate the reviewer's suggestion and have redrawn the cartoon model in Figure 7 to better illustrate the microtubule assembly process with Anna

Akhmanova and Michel O. Steinmetz's model¹ as an example.

Figure 7. Schematic illustration of the conformational changes of the MT assembly
 α - and β -tubulin are represented by green, blue (GTP bound) and pink (GDP bound) spheres. M-loops that are disordered and M-helices that are ordered are indicated by curved dashed lines and solid lines, respectively. Helix H7 and the intermediate domain (I) undergo a rotational movement during the curved-to-straight process, indicating different intra-dimer curvatures. By rotating the intermediate domain around Helix H7, the different Tube-bond and MT-bond interfaces are also reflected. In the process of assembling microtubules, two major steps occur sequentially: longitudinal contacts are formed first, followed by lateral contacts. In the early assembly stage, single-strand GTP-tubulin oligomers display less structural variation between different tubulin heterodimers than GDP-tubulin oligomers. Upon reaching the length of four heterodimers longitudinally, a single strand of GTP-tubulin starts to recruit new tubulin heterodimers and form lateral contacts containing three different states: the “encounter state” (one Tube-bond), the “transient state” (one MT-bond), and the “stable state” (two MT-bonds). Once a newly joined tubulin heterodimer (colored with dark blue and green) has gone through the whole process mentioned above, the intra-dimer curvature changes from $\sim 12^\circ$ to $\sim 0^\circ$.

Q3. The design of Movie S5 is very good for demonstrating the components packing purpose. However, some parts of this long movie can be more polished.

For example, at the 59", the movie may use a smoother transition so that readers understand that the component is placed back in the whole tube. There are several places in the movie that share a similar issue and can be polished.

R3. Thanks for the suggestion. We have revised the movie to make smooth transitions between different scenes.

Q4. There are previously published microtubule structural studies with some information about the intra-dimer interactions and/or inter-protofilament lateral interactions that mentioned salt bridge(s) like that reported in this manuscript. They seem to be neglected by this manuscript.

R4. We have reviewed the previously published microtubule structural studies. Consistent with these reported studies, the MT-bond interface mainly contains three major secondary structures, M-loop, H1-S2 and H2-S3 loop²⁻⁵. Unfortunately, the resolution of our current cryo-EM structures is not sufficient enough to discuss the detailed interaction such as salt bridges. We have included the related descriptions and citations in the results section of "The conversion from "Tube-bond" to "MT-bond"".

Response to Reviewer #3

This manuscript by Zhou et al. aims to uncover the mechanism behind the role of GTP binding and hydrolysis in microtubule (MT) assembly, a critical process in many eukaryotic cellular events. Through a series of cryo-electron microscopy structures and molecular dynamics simulations, the authors found differences in the flexibility of GDP- and GTP-bound tubulin heterodimers, and described sequential steps in the formation of lateral contacts during the early stages of MT assembly. They propose a flexible model of GTP-initiated MT assembly to explain the nucleation and assembly of MT. The findings are certainly important, as they offer new structural insight into the underlying mechanism of GTP binding and hydrolysis triggering MT assembly.

A major strength of the study is the use of single-particle cryo-electron microscopy (cryo-EM) to provide detailed tubulin heterodimer and oligomer structures as a function of nucleotide state. To the best of my knowledge, this is the first study that has resolved the in-solution structure of a single tubulin dimer without the addition of MT-depolymerizing drugs or co-factors (such as stathmin). The manuscript also markedly advances the field by describing the 3D structure of short tubulin oligomers, which offers a more complete picture of the assembly process than

previous 2D image analyses (e.g., Ayukawa et al. 2022). These oligomeric tubulin structures will also enable quite informative comparisons to future tubulin oligomers simulations. This is certainly a major breakthrough and should definitely be published.

However, I have some major issues, particularly regarding the second part of the manuscript, which need to be addressed before publication.

We thank the Reviewer for being enthusiastic about our study and for recognizing its novelty and significance. As described below, we have revised the manuscript to address all of the points raised in the review.

Q1. Extraction of flexibility from the observed three cryo-EM classes. The authors draw conclusions about the difference between the flexibility of GDP- and GTP-(analog)-tubulin tetramers from the 'variation range' observed in three cryo-EM classes each, without considering the different observed populations and without considering how structural classes that may not have been observed (e.g., in ED Fig 5 there is an additional 5% class that is disregarded) might affect the result. Also, it is unclear to what extent the used rigid body fitting of dimers might affect the result in comparison to flexible fitting (e.g., Rosetta, CDMD, MDFF), which this referee would think is more appropriate here. Also, the fits are ensemble averages and it is unclear whether the average over the full ensemble (as opposed to 3 class averaged structures) would yield similar results.

Given these uncertainties, it is hard to see why the difference between a range $[-44^\circ, -20^\circ]$ and $[-41^\circ, -24^\circ]$ should point to a significant difference in flexibility, particularly because much of the ranges are due to the least populated classes. Perhaps, as a better alternative, the authors might consider reporting the weighted standard deviation of the respective angles, which, for the radial bending, yields $\sim 7.7^\circ$ for GDP and $\sim 6.4^\circ$ for GMPCPP. To establish that this difference is indeed significant, the authors will need to provide a conservative error estimate for these angles. Also, this referee was unable to see why the combined angles lie outside the intervals defined by the respective three classes in most cases.

R1. We agree with the Reviewer that using SD values here is more accurate and reasonable to reflect the difference in flexibility, and we have incorporated the weighted standard deviation into Extended Data Table 1. The SD value for radial bending is 7.7° for GDP and 6.4° for GMPCPP; the SD value for tangential bending is 2.2° for GDP and 1.5° for GMPCPP; and the SD value for twist is 3.2° for GDP and 0.7° for GMPCPP. GDP-bound tubulin tetramer displayed a larger weighted standard deviation angle in every direction. This SD difference is therefore

confirmed in the GDP-tubulin tetramer, which exhibits a greater degree of flexibility. We added short descriptions at the end of the paragraph “To assess and better understand whether the nucleotide plays an important role in the inter-dimer interaction” in section “The inter-dimer interface of GDP-tubulin and GTP-tubulin tetramer” in the main text.

During data processing, we removed the class with the lowest particle number and the poorest map quality as a routine procedure in cryo-EM data processing to clean up the dataset for more reliable 3D classification. Because of the very small percentage of the removed particles, it is both necessary and reasonable to perform this step.

For the question of why the combined angles lie outside the intervals defined by the respective three classes, our explanation is as following. During the 3D classification step, different conformations were observed for GDP and GMPCPP bound tubulin tetramers. This indicates that tubulin tetramer conformations in solution are highly flexible and that tubulin tetramer particles are heterogeneous. To conduct a further 3D Variability Analysis in cryoSPARC, we combined particles from these different classes and ran an iteration of refinement. Based on the fact that we forced three different classes into one, the structure derived from the combined particles and the absolute bending angle value calculated from this structure are inaccurate. We have now replaced the angle calculated from the combined structure with the weighted average and standard deviation in Extended Data Table 1. Despite the fact that the angle calculated from combined structure itself is not accurate, the relative comparison between GDP and GMPCPP state does have some relevance. A large difference in the tangential bending angle of GDP-tubulin tetramer is a consequence of a larger tangential variation.

Q2. Interpretation of different flexibilities. The mechanism derived from these observations is probably not the only possible one. For instance, if GDP-oligomers are more flexible tangentially, shouldn't it be easier to interact with neighboring oligomers laterally? Accordingly, the combined data might as well suggest a different mechanism: because GDP-oligomers have, on average, a much larger tangential deformation than GMPCPP-oligomers, they are less likely to form a proto-lattice where all dimers could adopt a regular “MT-bond”.

R2. We agree with the Reviewer's opinion that there may be other interpretations. Our current model may be the most appropriate one. We believe that the stable lateral contacts are more difficult to establish than longitudinal contacts. As the reviewer pointed out, “if GDP-oligomers are more flexible tangentially, it might be easier to interact with neighboring oligomers laterally?” We think that it may be easier to collide with neighboring tubulin, but it might be more difficult to establish stable contacts. We appreciate and thank the reviewer for the comments regarding

"GDP-oligomers have, on average, a much larger tangential deformation than GMPCPP-oligomers, therefore, they are less likely to form a proto-lattice where all dimers would adopt a regular MT-bond." Our description has been revised, and we have now stated that either a greater degree of tangential flexibility, or a greater angle of tangential bending, or both may inhibit the formation of stable lateral contacts and prevent the assembly of microtubules.

Q3. Why do the authors provide no characterization of the observed tubulin hexamers? The hexamers encompass two inter-dimer bonds and their conformation would in theory be less affected by short-size artifacts. This is a valuable piece of information and could also serve to estimate the required uncertainties above.

R3. We agree with the Reviewer that hexamers contain more structural information. Since hexamer particles are not present in every micrograph, there are fewer hexamer particle images; additionally, hexamers are composed of two inter-dimers, which enhances structural flexibility. This has been verified by 2D classification results (shown below) that hexamers introduce more heterogeneous and complicated conformations. In addition, due to the small size of tubulin hexamer particles, it is not enough to obtain a reliable 3D reconstruction to proceed with further structural analysis.

Figure R3. The 2D classification result of tubulin hexamer particles

A. The 2D classification result of GDP state. B. The 2D classification result of GMPCPP state.

Q4. Section "The conversion from Tube-bond to MT-bond." The MD simulation part is based on the assumption that tubulins assembling into proto-lattice oligomers indeed undergo the observed conformational transition, which should be stated. Also, the authors should better explain exactly which question the simulations try to

address.

Q4.1 The authors claim that “the MD simulations demonstrate the initial formation of the Tube-bond followed by the conversion in the MT-bond.” In the supplementary material, they however write that “the potential energy” for their target structure (three laterally coupled tubulin dimers) “was defined to have a global minimum corresponding to the MT configuration.” It is not surprising that the authors observed the formation of, first, a Tube-bond between the three dimers followed by the formation of a MT-bond – indeed, the simulation was biased that way. So which additional information do these simulations provide in light of the main conclusions of this study? In my view, unless I overlooked something, the target-biased simulations could be omitted.

R4.1 We thank the reviewer for his or her comments to improve the readability of our work. For better understanding of the kinetic and dynamic property of the lateral interactions, we conducted structure-based MD simulations (with Gō model) starting from the solution tubulin structures to explore (1) the spontaneous assembly process from solution structure to final “MT-bond” interface; (2) whether the assembly process undergoes the direct “Tube-bond” to “MT-bond” conversion and (3) the key interactions between neighboring dimers that drive the lateral assembly process.

In general, current MD simulations can be divided into physics-based and structure-based (Gō like) models. The formers are usually based on an empirically parameterized force field and do not require prior knowledge of the simulating structure, while the structure-based model (SBM) is based on energy landscape theory and the principle of minimal frustration. The common feature of SBM is that some (or all) of the intra/inter-molecular interactions are explicitly defined to stabilize an experimentally determined structure. That is, rather than assigning energetic parameters solely on the basis of chemical composition (as done in physics-based models), potential energy minima are defined to stabilize known “native” conformations, and the long-timescale dynamics of biomolecules can be explored with low computational requirements. When applying SBM to assemblies, such as the ribosome^{6,7} or protein filaments^{8,9}, assigning experimentally determined structures (low free-energy states) to be potential energy minima to investigate the assembly process is quite common.

Therefore, it is rational for us to define the MT configuration as the potential energy minima and performed structure-based MD simulations starting from the solution tubulin structures to confirm whether the assembly process undergoes the “Tube-bond” interface and characterize the key interactions that drive the lateral assembly process. Also, assigning “MT-bond” state as the potential energy minima does not mean the assembly process must undergo the so-called intermediate

“Tube-bond” state, and vice versa. Actually, in our simulations setting “Tube-bond” state as the potential energy minima, very few MD trajectories would undergo the “MT-bond” state before reaching the targeted “Tube-bond” state (data not shown), supporting the direct transition from “Tube-bond” to “MT-bond” interfaces.

According to the reviewer’s suggestion, we have explained the questions we try to address and the major setup of the structure-based MD simulations in the revised manuscript.

Q4.2 The authors also used the MM-GBSA method to estimate the free energies of Tube- and MT-bonds. This is a quite inaccurate method, which should be clearly stated. In this context it would be good to compare/validate the obtained free energies results against values published previously using different methods?

R4.2 Thanks for the reviewer’s kind suggestions. The MM/GBSA method is widely used in the studies of the binding affinity and binding modes between small molecules¹⁰⁻¹² and protein or protein and protein¹³, which are in good agreement with the experimental results. In order to understand the strength of lateral interaction between the “MT-bond” and “Tube-bond” interfaces, we have performed MD simulations and analyzed the binding free energies of the two interfaces to validate the binding difference and state the dominated residues. Our simulation results reveal that the “MT-bond” interface is more stable than the “Tube-bond” interface, since its binding free energy is much stronger.

D. Tong and Gregory A. Voth¹⁴ used MD simulations to investigate the effects of different nucleotide states (GDP or GTP) on the conformational dynamic, and the results show that lateral interactions between neighboring protofilaments are significantly weakened for GDP state. They also provide the structural details for the lateral interacting tubulin domains at the seam interface of the α tubulin, and show the key interacting residue pairs (K124-D297 and D124-K338) are for the GTP state. Among the interacting residue pairs obtained from our calculations of MT-bond state, R123-E297 and E124-K338 were also found, showing good agreement with the work of D. Tong and Gregory A. Voth. Manandhar et al.¹⁵ studied the dynamic instability of different nucleotide states of the β subunit of the tubulin dimers, and their native contact analysis of lateral interface shows that the contacts are located in M loop, H1-S2 Loop, and H2-S3 Loop. The dominated residues obtain from our MM/GBSA analysis are also mostly located in this region.

Therefore, the simulation results of the lateral interface using different methods from previously published literatures are consistent with our MM/GBSA results.

Q5. Minor issues and readability for general audience:

Section “Capture of MT assembly intermediates from *Drosophila* S2 tubulin”.

The transition from bovine and porcine tubulin dimers and short oligomers to

resolving the structure of “GTPgammaS-Tubes” made of *Drosophila* tubulin is probably difficult to understand for the non-expert. In particular, the following questions may arise:

Q5.1 Why the switch of the tubulin species? If Tube and ribbon structures can't be observed with mammalian tubulin under the same conditions as for the oligomers, does this sufficiently justify the switch to insect tubulin?

R5.1 The tube and ribbon structures with mammalian tubulin were observed in a previous paper using non-hydrolyzed GTP analogue GMPCPP within 15-20 mM Mg^{2+} ¹⁶⁻¹⁸. However, the previous reconstruction is insufficient for obtaining high resolution structural details. Now we can assemble the tube and ribbon structures with S2 tubulin using non-hydrolyzed GTP analogue GTPγS within 2-5 mM Mg^{2+} , a magnesium concentration more close to the physiological levels. S2 tubulin is therefore used to assemble tube and ribbon structures with GTPγS. In addition, the current high resolution structure allows us to obtain more structural details. As Reviewer #4 suggested, we have also performed sequence alignment and structure alignment in Extended Data Figure 11F,G. The alignment demonstrated that both tubulin species share high conservation. To summarize, switching the tubulin species is reasonable and will not affect the final results.

Q5.2 If we do observe them for *Drosophila* tubulin and GTPgammaS, can we really conclude that these are indeed structural intermediates of MT assembly and not alternative and/or misassembled forms of *Drosophila* tubulin?

R5.2 In our view, tube and ribbon structures represent structural intermediates for the following reasons: 1. Helical ribbons and tubes exhibit a similar pattern to microtubule growth sheets. These growing sheets represent a transient structure during microtubule assembly; they are most likely in an intermediate GTP-Pi state, and GTPγS mimics this state¹⁹. 2. We have observed the transition from Tube to MT in the negative-staining samples. Therefore, we speculate that the Tube structure represents structural intermediates of the MT assembly process and not alternative and/or misassembled forms of *Drosophila* tubulin.

Q5.3 I understand that it is very hard to capture larger oligomers like laterally coupled hexamers, but isn't the step from a small oligomer to a tubulin-kinesin double-layer tube too large to draw conclusions about the initial steps of tubulin oligomerization?

R5.3 Microtubule assembly is a complicated process, probably involving many intermediate steps than we have discussed in this work. In this study, we focus on

the conformational changes of the tubulin heterodimer, since it is the smallest assembly unit. In order to investigate the mechanism of microtubule assembly, we compared the conformation of the minimum assembly unit of the tubulin heterodimer during microtubule assembly stages. Even though tubulin-kinesin double-layer tubes are very large, we focus on structural comparisons of tubulin heterodimer in the intermediate state of the Tube lattice and that of different assembly stages. Therefore, it would be reasonable to construct this stepwise microtubule assembly model based on a structural comparison of the tubulin heterodimer at various stages of assembly.

Q6. It would be useful to discuss how the results agree with recent simulations (e.g., Alexandrova et al, 2022. PNAS 119 (46) e2208294119, Igaev & Grubmüller, 2022. PNAS 119 (12) e21155161)?

R6. The contributions of Alexandrova et al. and Igaev & Grubmüller focus on the effect of the activation energy barriers on lateral interactions and microtubule assembly^{20,21}. The work of Igaev & Grubmüller showed that GTP binding at the interdimer interface causes structural changes of lateral contacts, hence raising the lower activation energy barriers than GDP-tubulin for straight lattice formation, which translates into its ability to elongate²⁰. Furthermore, the work of Alexandrova et al. constructed a four-state Monte Carlo model to confirm the activation energy barrier of lateral tubulin interactions can cause lagging curved protofilaments at the growing microtubule tips and thus destabilize the microtubule assembly²¹. These data are consistent with our experimental results that lateral contacts are critical components for microtubule assembly. Our cryo-EM data suggest that GDP binding increases tubulin inter-dimer flexibility, especially in tangential directions, preventing stable lateral contacts and microtubule assembly.

Q7. The methods part describing the MD-simulations lacks detail to an extent that the reported simulations are not reproducible. Examples are:

“Energy minimization was performed by imposing a strong restraint on each system”
– how strong precisely?

“[...] by minimizing the whole system for a few thousand steps.” – how many steps precisely? Which method? Conjugate gradient? Steepest descent?

“[...] by heating the whole system slowly from 100 to 300 K,” – How long? Linearly with time or exponentially by a weak T-coupling constant?

Which pressure coupling constant was used? Which pressure? (I guess 1 Atm, but should be specified.)

R7. Thanks the reviewer for the kind suggestion to make our MD simulations more reproducible. The missing simulation details have been supplemented to the “Molecular dynamic simulations” section of the updated manuscript:

“The entire systems were subject to the energy minimization in three stages to remove the bad contacts. Firstly, the solvent and the neutralized ions were minimized by holding the protein and ligand using a restraint with strength of 100 kcal/(mol Å²), and then the minimization was performed by holding the protein and ligand using a constraint of 10 kcal/(mol Å²). Finally, the whole systems were minimized by removing any constraint. Each stage was performed using the steepest descent minimization of 1000 steps followed by a 9000 steps conjugate gradient minimization. NVT (constant Number of atoms, Volume and Temperature) simulations were carried out by heating the whole system linearly with time gradually from 100 to 300 K in the first 300 ps, and the Berendsen thermostat was used to maintain the temperature of the whole system. Subsequently, the system was equilibrated under the temperature of 300 K for 1 ns was followed by a NPT (constant Number of atoms, Pressure, and Temperature) production run. During the heating stage, all the protein and ligands were restrained by a strength of 100 kcal/(mol Å²), and under equilibration stage the restraint strength was decreased to 10 kcal/(mol Å²). During the NPT production run, the Berendsen barostat was used to control the pressure at 1 atm, and the Langevin thermostat was employed to control the temperature of the system at 300K.”

Q8. Lines 92-106: The description of previous work needs improvement. For example, not all of the cited studies used MD simulations, contrary what is claimed in the text. The authors cite a very heterogeneous and somewhat unmotivated mix of papers that used phenomenological models of MT assembly, empirical mechanochemical models, coarse-grained MD, and unbiased all-atom MD as well as free energy calculations without clearly explaining the context. Also, the list of the cited simulation studies is missing some key contributions from the recent years.

R8. Thanks for the reviewer’s suggestion. As a result, we have revised the citations, removed the unrelated citation, added new simulation studies, and clarified some of the descriptions. Please refer to the “Introduction” section of our revised manuscript for details.

Response to Reviewer #4

Zhou et al’s manuscript addresses one of the fundamental questions in the cytoskeletal field, namely the mechanism of assembly of the microtubule polymer from soluble tubulin and how it is regulated by hydrolysis of bound GTP by the

β -tubulin subunit. Much work has already been done on this problem, but, here, the authors take several new approaches that substantially contribute to the field. The authors report four distinct types of structure using cryo-electron microscopy, including (for the first time) structures of porcine tubulin dimers in a solution state bound to either GDP or the slowly-hydrolyzing GTP analog, GMPCPP, as well as spontaneously forming small oligomers of tubulin dimers (tetramers and hexamers). This represents a true technical feat, due to the small size of these structures and low frequency of the oligomers. In addition, they report the first structures of *Drosophila melanogaster* tubulin assembled into microtubules, as well as an analog of a late-stage MT assembly intermediate. Combining this panoply of structures with modeling using molecular dynamics, they provide a novel perspective on how MTs assemble. This work will be of great interest to the readers of Nature Communications, and we recommend its acceptance.

We thank the Reviewer for being enthusiastic about our study and for recognizing its novelty and significance. To address all points raised in the review, we have revised the manuscript as described below.

Our main critical comment is that, to make the work more accessible to a general audience, the results should be more explicitly situated into the prior, extensive literature on microtubule structure and microtubule assembly. Below, we have indicated some key places where additional figures or discussion will strengthen the article:

Q1. The authors state that determining the structure of soluble GDP and GMPCPP-bound tubulin will yield a more accurate picture of the MT building block than the previously determined crystal structures, which may be artificially constrained by crystal contacts and the addition of polymerization-inhibiting drugs. Now that the authors have solved the structure of soluble tubulin, a short discussion should be included of what differences, if any, are observed between these new structures and the prior crystallographic structures (similar to the discussion for the tubulin tetramers in lines 171-173). In addition, a superposition of the prior crystallographic tubulin tetramer structure(s) onto the ensemble of new tubulin tetramer structures either in Figure 2, or in a supplement, would aid the reader substantially.

R1. Thanks for the Reviewer's suggestion. Based on our cryo-EM structures, both GDP and GMPCPP bound tubulin heterodimers adopt a similar curved conformation (Figures 1A-1D), indicating that nucleotide status does not influence structural differences significantly. Before obtaining the real in-solution tubulin structure, we could not exclude the possibility that crystal contacts and polymerization-inhibiting drugs might have an impact on the tubulin structure itself.

Afterwards, we compared our tubulin intra-dimer structure with those previously reported crystallographic structures and found that the whole tubulin exhibits a minor variation (Extended Data Figs. 4A,B). In addition, we have noticed that even the upper tubulin heterodimer is not exactly the same as the lower one in the crystal structure (PDBID:6tiy), as shown in the Extended Data Fig. 4A. This also indicates the existence of flexibility at the tubulin intra-dimer interface. There will be a very small variation in the tubulin heterodimer structure as a result of the intra-dimer flexibility. Thus, we consider these minor structural vibrations not to indicate significant structural differences, instead to indicate a minor structural flexibility within the intra-dimer interface. We have added a description to the main text of the section titled “Structures of GDP-tubulin and GMPCPP-tubulin heterodimer in solution”.

For tubulin tetramers, the inter-dimer flexibility is greater and we were able to obtain three distinct conformations through cryo-EM rather than a fixed conformation based on X-ray crystallography. The superposition of these tubulin tetramers has now been added as Extended Data Fig. 4C, which we hope will be useful to the reader. We now have added a small discussion in the main text on the section of “The flexibility of intra- and inter-dimer interface”.

Q2. The presence of tubulin tetramers and tubulin hexamers in both GDP and GMPCPP samples is very interesting and the comparison between the two structural ensembles is well done. It would be beneficial to the reader to understand how the frequency of tubulin tetramer and tubulin hexamer compares between the GDP and GMPCPP samples. Can the authors quantify the average number of tetramer particles per micrograph as a percentage of total particles or as a ratio with tubulin dimers? This is valuable information that would speak to the authors’ model and could help substantiate the claim in the discussion (lines 404-405) that GDP-tubulin is unfavorable to initiate MT assembly due to a larger range of motion in the tetramer, and not because formation of tetramers themselves is disfavored. The number of hexamer particles should also be quantified in the same way to provide a further data point. This information could be easily incorporated into Extended Data figures 1 and 2.

R2. We appreciate the Reviewer’s suggestion. As hexamer particles are rare and do not appear on every micrograph, we traced back raw micrographs containing hexamer particles based on their 2D classification results. We then selected 50 micrographs at random from those containing hexamer particles. The GDP state dataset contains 4,252 tubulin heterodimer particles, 2,222 tubulin tetramer particles, and 379 tubulin hexamer particles. As a percentage of the total particles per micrograph, tubulin dimers, tetramers, and hexamers account for 62%, 32%, and 6%, respectively. According to the GMPCPP dataset, there are 4,498 tubulin

heterodimers, 1,990 tubulin tetramers, and 472 tubulin hexamers. The proportion of tubulin dimers, tetramers, and hexamers in each micrograph is 64%, 29%, and 7%. According to the results, the nucleotide status does not induce a shift or increment in the tetramer/dimer ratio. These data further confirm our model and demonstrate that GDP-tubulin is unfavorable to initiate MT assembly due to a larger range of motion in the tetramer, but not due to the disfavored formation of tetramers themselves. We have included these results in Extended Data Fig. 2C and a short description in the discussion section of the main text entitled "The flexibility of intra- and inter-dimer interfaces". We also added a small paragraph of explanation to the method section.

Q3. The use of *Drosophila melanogaster* tubulin is a clever strategy to circumvent temperature and kinetic limitations inherent to the standard mammalian tubulins. However, it raises the issue of whether results using *D. melanogaster* tubulin can be extrapolated to mammalian tubulin. At minimum, the authors should include the sequence identity between bovine and *D. melanogaster* tubulin in the main text (most likely in the Results subsection 'Capture of MT assembly intermediates from *Drosophila* S2 tubulin'), and also a sequence alignment, including all species mentioned in the Discussion paragraph 'Tubulin mutations and their associated abnormal phenotypes' and highlighting the amino acids found within the 'Tube-bond' and 'MT-bond.' This paragraph in the discussion should also either reference an existing figure or be accompanied by a new figure that highlights the residues mentioned.

R3. Thanks for the Reviewer's suggestion. Sequence alignment has been performed and the result has been presented as Extended Data Fig. 17. In the discussion section of "Tubulin mutations and their associated abnormal phenotypes", we have made a brief mention of sequence alignments and referred to this figure. Sequence alignment results indicate that all tubulin sequences mentioned are highly conservative. Our structural alignment results in the Extended Data Figs. 11F,G also indicate that their structure is highly conservative. With a non-hydrolyzed analogue of GTP and higher concentrations of Mg^{2+} , we observe similar tube and ribbon structures using *D. melanogaster* and porcine tubulin. We think results using *D. melanogaster* tubulin can be extrapolated to mammalian tubulin.

Extended Data Figure 17. Sequence alignment results among different species

A. α -tubulin sequence alignment result. OS is an abbreviation for *O. sativa* and HS is an abbreviation for *H. sapiens*. A blue box represents residues involved in Tube-bond, a red box represents residues involved in MT-bond, and a black box represents residues involved in both MT-bond and Tube-bond (same for B).

B. β -tubulin sequence alignment result. SC is an abbreviation for *S. cerevisiae*. AT is an abbreviation for *A. thaliana*.

Q4. The use of the kinesin-1 motor domain to improve and interpret the structure of the GTP γ S D. melanogaster tube is likewise very clever. However, given prior reports that addition of microtubule-binding proteins to D. melanogaster tubulin can alter assembly reactions, we were concerned that the kinesin motor domain might be influencing the resulting EM reconstruction, and thus that the kinesin-bound structure might not represent a true on-pathway assembly intermediate. Could the authors provide additional data that would clarify this point? The easiest option might be an overlay of the lower-resolution undecorated GTP γ S-Tube with the decorated higher resolution tube to show that the two structures are analogous. In addition, it would be beneficial to include a comparison between the structure of the previously published GMPCPP bovine tube (citation 46) within a figure. Note that this reference should be cited also at line 282.

R4. Unfortunately, we didn't obtain a low-resolution reconstruction of undecorated GTP γ S-Tube because we could hardly determine accurate helical symmetry with

the undecorated sample. However, there are two protofilaments in the GTPyS-tube, one is bound to kinesin, while the other is not bound to kinesin and its structure is not affected by kinesin. Additionally, according to the literature, “Kinesin binding has a small effect on the extended, GMPCPP-bound lattice, but hardly affects the compacted GDP-MT lattice. We find that kinesin-1 binding shortens the spacing between longitudinally interacting tubulin dimers (the dimer rise) of the GMPCPP-MT by 0.7 Å. On the other hand, kinesin-1 binding hardly affects the lattice spacing of the already compacted GDP-MT.” Kinesin has a small structural effect on the lattice, less than 1 Å. Whereas in our Figure 5E and 6E, the displacements are around 10 Å and 5 Å respectively. Because the displacement is substantially higher than the previously reported value affected by kinesin, we believe that kinesin is not responsible for these structural differences. Therefore, we think our structure could represent a true on-pathway assembly intermediate.

We followed the reviewer’s suggestion to compare the structures of GTPyS-Tube and GMPCPP-Tube. The MT-bond interface is similar, while the Tube-bond interface differs (Extended Data Figs. 16 A-D). According to our structural and MD data, M-loop, H10-S9 loop, H9-S8 loop, H6 helix, H9 helix and H10 helix of one tubulin interact with the neighbouring H3 helix, H4 helix, H5 helix and H4-S5 loop (Figures 3F and 4A; Extended Data Figs. 16 A-D; Extended Data Table 2). According to the reported GMPCPP-tube structure, H10-S9 loop and H9-S8 loop of one tubulin interact with H3 helix, H4 helix and H4-S5 loop of the neighboring tubulin¹⁶. There is substantial overlap between these two Tube-bond interfaces. However, the Tube-bond interface of GTPyS-Tube has a larger buried area (468 Å²) than that of GMPCPP-Tube (175 Å²). One side of tubulin heterodimer in the GMPCPP-Tube interface has shifted about 20 Å related to that in the GTPyS-Tube bond interface (Extended Data Fig. 16D). We consider that different tube-bonds may correspond to different intermediate states of lateral contacts. It is possible, for example, that the GMPCPP state mimics the very beginning state, followed by the GTPyS state. Due to the differences in lateral contacts, the tubulin dimer structure has minor structural differences.

We have added a paragraph about the structural comparison between GTPyS-Tube and GMPCPP-Tube in the discuss section of “Lateral interaction changes the conformation of tubulin heterodimer”. And we have cited the reference at line 282.

Extended Data Figure 16. The structural comparison between GTP γ S-Tube and GMPCPP-Tube

(A-B) Structural comparison of “MT-bond” interface between GTP γ S-Tube and GMPCPP-Tube. (A) Side view. (B) Top view. Tubulin structures are superimposed on the right tubulin heterodimer of “MT-bond” interface. The left tubulin heterodimer of “MT-bond” (the right tubulin heterodimer of “Tube-bond” interface) of GTP γ S-Tube is colored in purple and the right one is colored in orange. The left tubulin heterodimer of “MT-bond” (the right tubulin heterodimer of “Tube-bond” interface) of GMPCPP-Tube is colored in aquamarine and the right one is colored in tan. (same for (C-F))

(C-D) Structural comparison of “Tube-bond” interface between GTP γ S-Tube and GMPCPP-Tube. (C) Side view. (D) Top view. Tubulin structures are superimposed on the right tubulin heterodimer of “Tube-bond” interface.

(E-F) Structural comparison of tubulin heterodimers in the GTP γ S-Tube and GMPCPP-Tube lattice, superimposed on α -tubulin.

Q5. The distribution of protofilament number among microtubules for the *D. melanogaster* GTP γ S-kinesin-1 structure is different from mammalian GTP γ S microtubules, in that it is dominated by 15-protofilament microtubules. Mammalian GTP γ S microtubules, however, are predominantly 14-protofilament (Zhang R, LeFrance B, Nogales E. PNAS 2018). The authors should comment on this, as it may be of interest to the readers.

R5. We compared the reconstitution method with the reference²². We did not use GMPCPP-seeds in our experiment. The s2-MT was assembled in the presence of 1.5 mM GTPγS and 5% DMSO (vol/vol) as described in the method. It is possible that using GMPCPP-seed will affect the number of protofilaments.

Minor comments:

Q6. The comparison of bovine and *Drosophila* tubulin dynamics (Extended Data Figure 13) is not referenced or discussed in the text. This could be added, eg, to the Results section around line 224.

R6. The reviewer may miss this. In the previous version, Extended Data Fig. 13 is referenced in Line 412. “This is probably due to the natural selection of insect tubulin with higher polymerization property at relatively low temperature (Extended Data Fig. 13).”

Q7. The temperature the dynamic assays were performed at, as well as other key assay conditions including Mg^{2+} concentration, glycerol, buffer, etc, should be included in the Methods section and, potentially, in the legend of Extended Data Figure 13 (eg bovine tubulin polymerizes at a different optimal temperature than *D. melanogaster* tubulin, which does not make bovine tubulin intrinsically less capable of polymerizing). Also, it should be clarified whether the seeds the soluble tubulin is polymerizing from are bovine or *Drosophila*.

R7. We have supplemented the detailed information in the method of microtubule dynamic assay. “Briefly, GMPCPP microtubules (5% Alexa Fluor 647 labeled and 20% biotin labeled) were stabilized on the surface of the cover glass coated with a biotin-binding protein. Porcine tubulin and *Drosophila* S2-tubulin were used to polymerize GMPCPP microtubules in the porcine and S2 microtubule dynamics assays, respectively. Dynamic microtubules started to grow from GMPCPP microtubules under 35°C when porcine tubulin or *Drosophila* S2-tubulin was added to the flow cell. BRB80 supplemented with 2 mM GTP, 50 mM KCl, 0.15% sodium carboxymethylcellulose, 80 mM D-glucose, 0.4 mg/ml glucose oxidase, 0.2 mg/ml catalase, 0.8 mg/ml casein, 1% β-mercaptoethanol, 0.001% Tween 20 was used as the imaging buffer in our microtubule dynamic assay. The dynamics of microtubules was recorded by a total internal reflection (TIRF) microscope (Olympus) equipped with an Andor 897 Ultra EMCCD camera (Andor, Belfast, UK) using a 100× TIRF objective (NA 1.49; Olympus).”

Q8. Dithiothreitol (DTT) should be spelled out (line 878) the first time it is mentioned.

R8. Thanks for the Reviewer's remind. We have replaced DTT with Dithiothreitol.

Q9. The versions of Fiji (line 890), Topaz (line 952), COOT (line 1001), PHENIX (line 1002), and Chimera (line 1004) should be specified.

R9. We have supplemented the version information in the methods section.

Q10. In the discussion section 'A hypothetical model of MT assembly', the authors claim at line 476 that 'The statistics for negative staining EM data shows that...'. Are these statistics coming from citation 28? If so, that should be clarified.

R10. Yes, these statistics come from citation 28. We have clarified this by adding the citation 28 at the end of this sentence.

Q11. The paper will need to be generally proof-read for minor typos and grammar issues prior to publication

R11. Thanks for the Reviewer's suggestion. We have carefully proofread our manuscript.

References

- 1 Akhmanova, A. & Steinmetz, M. O. Control of microtubule organization and dynamics: two ends in the limelight. *Nature reviews. Molecular cell biology* **16**, 711-726, doi:10.1038/nrm4084 (2015).
- 2 Sui, H. & Downing, K. H. Structural basis of interprotofilament interaction and lateral deformation of microtubules. *Structure (London, England : 1993)* **18**, 1022-1031, doi:10.1016/j.str.2010.05.010 (2010).
- 3 Zhang, R., Alushin, G. M., Brown, A. & Nogales, E. Mechanistic Origin of Microtubule Dynamic Instability and Its Modulation by EB Proteins. *Cell* **162**, 849-859, doi:10.1016/j.cell.2015.07.012 (2015).
- 4 Alushin, G. M. *et al.* High-resolution microtubule structures reveal the structural transitions in $\alpha\beta$ -tubulin upon GTP hydrolysis. *Cell* **157**, 1117-1129, doi:10.1016/j.cell.2014.03.053 (2014).
- 5 Manka, S. W. & Moores, C. A. The role of tubulin-tubulin lattice contacts in the mechanism of microtubule dynamic instability. *Nature structural & molecular biology* **25**, 607-615, doi:10.1038/s41594-018-0087-8 (2018).
- 6 Levi, M., Walak, K., Wang, A., Mohanty, U. & Whitford, P. C. A steric gate controls P/E hybrid-state formation of tRNA on the ribosome. *Nature communications* **11**, 5706, doi:10.1038/s41467-020-19450-0 (2020).
- 7 Nguyen, K. & Whitford, P. C. Steric interactions lead to collective tilting motion in the ribosome during mRNA-tRNA translocation. *Nature communications* **7**, 10586, doi:10.1038/ncomms10586 (2016).
- 8 Faelber, K. *et al.* Structure and assembly of the mitochondrial membrane remodelling GTPase Mgm1. *Nature* **571**, 429-433, doi:10.1038/s41586-019-1372-3 (2019).
- 9 Noel, J. K., Noé, F., Daumke, O. & Mikhailov, A. S. Polymer-like Model to Study the Dynamics of Dynamin Filaments on Deformable Membrane Tubes. *Biophysical journal* **117**, 1870-1891, doi:10.1016/j.bpj.2019.09.042 (2019).
- 10 Plenge, P. *et al.* The mechanism of a high-affinity allosteric inhibitor of the serotonin transporter. *Nature Communications* **11**, 1491, doi:10.1038/s41467-020-15292-y (2020).
- 11 Jansen, J. A. *et al.* Structural basis for proficient oxidized ribonucleotide insertion in double strand break repair. *Nature Communications* **12**, 5055, doi:10.1038/s41467-021-24486-x (2021).
- 12 Wang, H. *et al.* A structural exposé of noncanonical molecular reactivity within the protein tyrosine phosphatase WPD loop. *Nature Communications* **13**, 2231, doi:10.1038/s41467-022-29673-y (2022).
- 13 Kotila, T. *et al.* Mechanism of synergistic actin filament pointed end depolymerization by cyclase-associated protein and cofilin. *Nature Communications* **10**, 5320, doi:10.1038/s41467-019-13213-2 (2019).
- 14 Tong, D. & Voth, G. A. Microtubule Simulations Provide Insight into the Molecular Mechanism Underlying Dynamic Instability. *Biophysical journal* **118**, 2938-2951, doi:10.1016/j.bpj.2020.04.028 (2020).
- 15 Manandhar, A., Kang, M., Chakraborty, K. & Loverde, S. M. Effect of Nucleotide State on the

-
- Protofilament Conformation of Tubulin Octamers. *The journal of physical chemistry. B* **122**, 6164-6178, doi:10.1021/acs.jpcc.8b02193 (2018).
- 16 Wang, H. W. & Nogales, E. Nucleotide-dependent bending flexibility of tubulin regulates microtubule assembly. *Nature* **435**, 911-915, doi:10.1038/nature03606 (2005).
- 17 Wang, H. W., Long, S., Finley, K. R. & Nogales, E. Assembly of GMPCPP-bound tubulin into helical ribbons and tubes and effect of colchicine. *Cell cycle (Georgetown, Tex.)* **4**, 1157-1160, doi:10.4161/cc.4.9.2042 (2005).
- 18 Nogales, E. & Wang, H. W. Structural intermediates in microtubule assembly and disassembly: how and why? *Current opinion in cell biology* **18**, 179-184, doi:10.1016/j.ccb.2006.02.009 (2006).
- 19 Guesdon, A. *et al.* EB1 interacts with outwardly curved and straight regions of the microtubule lattice. *Nature cell biology* **18**, 1102-1108, doi:10.1038/ncb3412 (2016).
- 20 Igaev, M. & Grubmüller, H. Bending-torsional elasticity and energetics of the plus-end microtubule tip. *Proceedings of the National Academy of Sciences of the United States of America* **119**, e2115516119, doi:10.1073/pnas.2115516119 (2022).
- 21 Alexandrova, V. V. *et al.* Theory of tip structure-dependent microtubule catastrophes and damage-induced microtubule rescues. *Proceedings of the National Academy of Sciences of the United States of America* **119**, e2208294119, doi:10.1073/pnas.2208294119 (2022).
- 22 Wang, A., Zhang, Z. & Li, G. Higher Accuracy Achieved in the Simulations of Protein Structure Refinement, Protein Folding, and Intrinsically Disordered Proteins Using Polarizable Force Fields. *Journal of Physical Chemistry Letters* **9**, 7110-7116, doi:10.1021/acs.jpcclett.8b03471 (2018).

REVIEWER COMMENTS

Reviewer #3 (Remarks to the Author):

The authors addressed some of our concerns, but quite a number of major concerns remain unaddressed. This is a bit unfortunate, as this referee really likes the work and would like to see it eventually published -- but without still remaining flaws and gaps in the arguments. The first reply 'R1' is a good and representative example, but the pattern of, in my view, too superficial polishing without changing the actual underlying issue in the manuscript continues further below. Therefore, it would be inappropriate to publish the manuscript in its present form, and I am happy to consider one more revision. I am convinced all issues can be addressed properly, and they should.

R1. The authors recognised the importance of weighted averages and standard deviations and adopted our proposal to recalculate the angular parameters (Extended Data Table 1). However, the main issue that triggered this remark has not been addressed, which is: do the observed angular differences b/w the GDP- and GMPCPP-tetramer conformations really support the claim that GDP vs GMPCPP affects the flexibility, i.e., are the observed differences statistically significant? To me, this question has not been satisfactorily answered, neither in the replies, nor in the text.

Related to this question is: How does rigid body fitting of dimer structures into the densities affect the results? Obviously, this question addresses possible systematic uncertainties on top of the above statistical uncertainties, which, in order to establish that the data provides sufficient evidence for actual differences in flexibility, must also be addressed. Unfortunately, this issue still seems unanswered, too.

My tentative interpretation of the above Table would be that GDP- and GMPCPP-tetramers differ mainly in tangential and twist deformations and not, as claimed, in radial bending (large SDs). GDP-tetramers appear to be more deformed tangentially but less twisted (although flexible), whereas GMPCPP-tetramers appear to be more twisted (and rigid) but less deformed tangentially. I think the authors should better explain the results of the above Table in the main text. Actually, in my opinion the Table is important enough to show it in the main text; I'll leave it to the authors to consider this possibility.

Concerning the question of why the combined angles lie outside of the angle ranges, I have to admit that I did not understand the authors' reply, even after triple reading. This issue thus remains to be properly addressed.

R2. Here the authors seem to acknowledge my point, but, unexpectedly to me, in the changes to the main text they seem to claim the opposite. To clarify: My point was that a larger tangential bending

flexibility should make lateral contact formation easier, to which the authors seem to agree. However, the authors now also wrote: "Either a greater degree of tangential flexibility [...] may inhibit the formation of stable lateral contacts and prevent the assembly of microtubules" – which is just the opposite. The authors should avoid such inconsistency in their own line of arguments.

R3. Fair answer. I would encourage the authors to also add this point to the manuscript, as this referee will certainly not be the only reader wondering why no characterisation of the hexamers has been performed.

R4.1 Here the authors missed my point, which was that the authors' conclusion "the MD simulations demonstrate the initial formation of the Tube-bond followed by the conversion in the MT-bond." is invalid. The reason why it is invalid is that this result was already built into the simulations, because a structure-based model (the Go model) was used that has been biased towards the desired result: If you pre-define the energy landscape in a simulation towards a certain structure (as opposed to a purely physics-based model), it comes as no surprise that the simulation produces this particular structure as the expected result. By the same token, any observed intermediates are questionable, because they might not reflect the true balance between the free energies of formed vs. not yet formed interactions. In this particular case, free dimers formed lateral bonds because the Go model was parametrised using end state of a trimer with two lateral bonds.

Note that this referee did not argue against the technical soundness of these simulations, as the authors seem to think. Rather, I challenged the conclusion, and rather than improving the readability, here that authors need to establish either that (and why) my above argument against their conclusion is invalid, or remove their conclusion altogether. My preference would obviously be the latter, because in my view the experimental evidence is already good enough, (provided that the assembly intermediates shown are not artefacts) and the simulations do not add any new information.

R4.2 Just because a method is widely used, doesn't imply that it's accurate enough for the particular question at hand. Although MM/GBSA did show some agreement in a number of studies, the authors ignore an even larger body of studies that put the accuracy of this method into question. The authors actually seem to share this view, because in their reply they resort to results by Tong & Voth as support for their MM/GBSA result. Unfortunately, the current literature also offers results that disagree with the MM/GBSA results, e.g., Igaev & Grubmuller, which the authors do cite but not here. To selectively cite literature that agrees without mentioning other literature that disagrees is not helpful. Again, this referee does not really see the point of including these simulation results within the paper, as the experiments are already sufficiently conclusive.

R5.1, R5.2, and R5.3 Thank you for the explanations, to which I fully agree. As above, the authors may wish to add a summary of it to the manuscript, as also other readers may wonder.

R6. I largely agree with the authors' reply, but apparently no change to the manuscript was made.

R7. The methods part describing the MD simulations is now sufficiently detailed to render the simulations reproducible, as requested.

R8. The description of previous work has been improved satisfactorily.

Response to reviewer #3

The authors addressed some of our concerns, but quite a number of major concerns remain unaddressed. This is a bit unfortunate, as this referee really likes the work and would like to see it eventually published -- but without still remaining flaws and gaps in the arguments. The first reply 'R1' is a good and representative example, but the pattern of, in my view, too superficial polishing without changing the actual underlying issue in the manuscript continues further below. Therefore, it would be inappropriate to publish the manuscript in its present form, and I am happy to consider one more revision. I am convinced all issues can be addressed properly, and they should.

R1. The authors recognised the importance of weighted averages and standard deviations and adopted our proposal to recalculate the angular parameters (Extended Data Table 1). However, the main issue that triggered this remark has not been addressed, which is: do the observed angular differences b/w the GDP- and GMPCPP-tetramer conformations really support the claim that GDP vs GMPCPP affects the flexibility, i.e., are the observed differences statistically significant? To me, this question has not been satisfactorily answered, neither in the replies, nor in the text.

Related to this question is: How does rigid body fitting of dimer structures into the densities affect the results? Obviously, this question addresses possible systematic uncertainties on top of the above statistical uncertainties, which, in order to establish that the data provides sufficient evidence for actual differences in flexibility, must also be addressed. Unfortunately, this issue still seems unanswered, too.

My tentative interpretation of the above Table would be that GDP- and GMPCPP-tetramers differ mainly in tangential and twist deformations and not, as claimed, in radial bending (large SDs). GDP-tetramers appear to be more deformed tangentially but less twisted (although flexible), whereas GMPCPP-tetramers appear to be more twisted (and rigid) but less deformed tangentially. I think the authors should better explain the results of the above Table in the main text. Actually, in my opinion the Table is important enough to show it in the main text; I'll leave it to the authors to consider this possibility.

Concerning the question of why the combined angles lie outside of the angle ranges, I have to admit that I did not understand the authors' reply, even after triple reading. This issue thus remains to be properly addressed.

Response to R1: We appreciate greatly the Reviewer's comments on the statistical analysis in our work. We have followed the reviewer's suggestion to perform more rigorous analysis of our data as stated below from different points of view.

As for the rigid body fitting, the reviewer stated in the last version of comments, "it is unclear to what extent the used rigid body fitting of dimers might affect the result in comparison to flexible fitting (e.g., Rosetta, CDMD, MDFF), which this referee would think is more appropriate here." Regarding the reviewer's comment about the use of rigid body fitting for dimers, flexible fitting

methods such as Rosetta, CDMD, and MDFF are more appropriate for higher resolution cryo-EM density maps with resolutions of at least 4-5 Å. At lower resolutions, the density map may not provide sufficient information about the conformational changes that occur during the fitting process, which can lead to inaccurate results. In this regard, the map resolution of our tetramer is more suitable for rigid body fitting (Extended Data Figs. 6 and 7). To address the potential error range of rigid body fitting on the angle measure of low resolution maps, we performed additional control test by lowpass filtering our high resolution cryo-EM map of MT to 8 Å, which is similar to the resolution we used in tubulin tetramer analysis. We performed rigid body fitting of tubulin monomer into the map and calculated the radial, tangential, and twist angles of intra-dimer. The resulting angles were -0.3, 0.8, and 0.2 degrees, respectively. We also calculated the angles from our atomic model derived from our high-resolution reconstruction of MT, which were -0.4, 0.6, and 0.2 degrees (Extended Data Table 3). This control test demonstrates that the angle measurement based on rigid body fitting in the map of 8 Å resolution is reasonably accurate with errors less than 0.2 degrees.

For the problem of the combined angle lying outside of the angle ranges, we found that it was due to the inappropriate use of the 3D variability results in our calculation. We therefore removed the combined angle analysis. Instead, we performed a more rigorous statistical analysis of the structures and now replace Fig. 2I-K by violin plots calculated from data in Extended Data Table 1 to address the reviewer's comments on statistical significance between GDP- and GMPCPP-tetramer angles. Considering the non-normal distribution of both datasets and the limited number of values (only three) within each dataset, we expanded our datasets by creating two new datasets based on the existing samples. The objective was to conduct a hypothetical test while ensuring the newly generated datasets to maintain the same mean and variance as the original datasets but with a larger number of samples. This expansion aimed to provide a more comprehensive dataset, facilitating a more robust statistical analysis. To generate the expanded dataset, we transformed the percentage values into integers by multiplying them by 100. For example, in the Tangential bending angle dataset, the GDP state now consisted of 42 samples of Class-2, 41 samples of Class-1, and 12 samples of Class-4. Likewise, the GMPCPP state encompassed 53 samples of Class-4, 29 samples of Class-2, and 16 samples of Class-3. To assess the statistical significance of these datasets, we performed a two-tailed U-test, specifically the Mann-Whitney U-test or Mann-Whitney-Wilcoxon test. This test was selected due to their suitability for nonparametric datasets, eliminating the need for assumptions of normal distribution or knowledge of distribution parameters. The U-test and corresponding p-values for each case, as illustrated in Figure 2 I-K, were calculated by Scipy toolbox¹ in Python³². The resulting p-values provide statistical evidence that the observed differences in the angular parameters between GDP- and GMPCPP-tetramers are indeed significant and support our claim that GDP vs GMPCPP affects the flexibility. We have incorporated the results of this analysis into the results section titled "The inter-dimer interface of GDP-tubulin and GTP-tubulin tetramer". Additionally, we have provided a description of the analysis process in the Methods section.

Figure 1 The illustration of generating new datasets of sample for U-test hypothesis test on the tangential bending (TB)

Two steps are involved in statistical significance analysis: the generation of new datasets and the performance of U tests on the new datasets. Boxplots display the mean angle value on the y-axis and the standard deviation on the segment length. For all panels, statistically significant results from two-sided U-tests are denoted as: $p\text{-value} < 0.05$; * $p\text{-value} < 0.01$; *** $p\text{-value} < 0.001$.

The reviewer stated “My tentative interpretation of the above Table would be that GDP- and GMPCPP-tetramers differ mainly in tangential and twist deformations and not, as claimed, in radial bending (large SDs)”. Our analysis, however, indicates that the standard deviation (SD) values of the GDP state are higher than those of the GMPCPP state in each direction. We did not claim that the difference is limited to radial bending. Furthermore, our three-dimensional variability analysis (3DVA) unveiled two prominent bending directions: radial and tangential. We observed a significant difference in tangential bending rather than radial bending. This distinction is evident when reviewing movies S1 and S3, where both GDP tubulin and GMPCPP tubulin display greater radial bending flexibility. However, only GDP tubulin exhibits greater tangential bending flexibility, as seen in movies S2 and S4. Based on these findings, we consider the tangential bend to be the key factor influencing microtubule assembly. To address the clarity of our main text, we have made appropriate edits to ensure that this point is more explicitly conveyed.

Figure 2. The curvature fluctuation around the inter-dimer interfaces of tubulin tetramer in solution

(A) Overview of three major conformations of GDP- and GMPCPP-tubulin tetramers. All models are aligned together using $\beta 1$ -tubulin as reference. GDP-tubulin tetramers are colored with coral, plum and khaki, respectively. GMPCPP-tubulin tetramers are colored with steel blue, light sea green and light grey, respectively.

(B) The coordinate system is defined to describe the bending angles of $\alpha 2$, $\beta 2$ -tubulin relative to $\alpha 1$, $\beta 1$ -tubulin. And the quantitative values of the angles are decomposed in the x (radial bending), y (tangential bending) and z (twist) axes.

(C-H) Bending of inter-dimer in radial, tangential and twist directions. The upper heterodimer ($\alpha 2$, $\beta 2$ -tubulin) shown in (A) of GDP-state (C-E) and GMPCPP-state (F-H). The black arrow indicates the direction of bending. A simplified cartoon model and its bending angle range are displayed in the left-bottom corner. H11 and H12 helices are represented as cylinders, the N and I domain are shown as surfaces.

(I-K) Violin plots represent the distributions of bending angle variations with their weights in each state for a two-sided U-test. The angle values are indicated by circles in each related violin plot. The boxplots represent the distribution of bending angle variations from three different directions. In both of the violin plots and the boxplots, the vertical axis indicates the bending angle value while the horizontal axis corresponds to different nucleotide states. Boxplots display the mean angle value on the y-axis and the standard deviation on the segment length. For all panels, statistically significant results from two-sided U-tests are denoted as: *p-value* < 0.05; * *p*-

value < 0.01; ***p-value < 0.001. (I) The distribution of radial bending angles. (J) The distribution of tangential bending angles. (K) The distribution of twist angles.

R2. Here the authors seem to acknowledge my point, but, unexpectedly to me, in the changes to the main text they seem to claim the opposite. To clarify: My point was that a larger tangential bending flexibility should make lateral contact formation easier, to which the authors seem to agree. However, the authors now also wrote: "Either a greater degree of tangential flexibility [...] may inhibit the formation of stable lateral contacts and prevent the assembly of microtubules" – which is just the opposite. The authors should avoid such inconsistency in their own line of arguments.

Response to R2: We agree that a larger tangential bending flexibility should make lateral contact formation easier to some extent, but it does not necessarily guarantee the formation of **stable** lateral contacts. While flexibility leads more opportunities for proteins to meet each other, the contacts however are not necessarily the favored stable interactions and they may dissociate shortly after collision. In contrast, the relatively rigid conformation of GMPCPP-tubulin is more favorable to the stable lateral interactions. To further clarify, the interface area for tubulin lateral contacts is about 468 Å², which contains multiple key residues interacting with each other (Figs. 3F and 4A). We have added the above statement in the discussion to make our points clearer.

R3. Fair answer. I would encourage the authors to also add this point to the manuscript, as this referee will certainly not be the only reader wondering why no characterisation of the hexamers has been performed.

Response to R3: Thanks for this suggestion. We have included the hexamer information in Extended Data Figure 8 and added a brief explanation at the end of the section titled "The inter-dimer interface of GDP-tubulin and GTP-tubulin tetramer" in the main text.

R4.1 Here the authors missed my point, which was that the authors' conclusion "the MD simulations demonstrate the initial formation of the Tube-bond followed by the conversion in the MT-bond." is invalid. The reason why it is invalid is that this result was already built into the simulations, because a structure-based model (the Go model) was used that has been biased towards the desired result: If you pre-define the energy landscape in a simulation towards a certain structure (as opposed to a purely physics-based model), it comes as no surprise that the simulation produces this particular structure as the expected result. By the same token, any observed intermediates are questionable, because they might not reflect the true balance between the free energies of formed vs. not yet formed interactions. In this particular case, free dimers formed lateral bonds because the Go model was parametrised using end state of a trimer with two lateral bonds.

Note that this referee did not argue against the technical soundness of these simulations, as the authors seem to think. Rather, I challenged the conclusion, and rather than improving the

readability, here that authors need to establish either that (and why) my above argument against their conclusion is invalid, or remove their conclusion altogether. My preference would obviously be the latter, because in my view the experimental evidence is already good enough, (provided that the assembly intermediates shown are not artefacts) and the simulations do not add any new information.

Response to R4.1: We thank the reviewer for the kind comments to improve our manuscript. Based on the analysis of EM results, also see R5.2 in the last revision, we propose that the tube and ribbon structures represent structural intermediates, then we intend to verify our hypothesis by using MD simulations.

“Q5.2 If we do observe them for *Drosophila* tubulin and GTP γ S, can we really conclude that these are indeed structural intermediates of MT assembly and not alternative and/or misassembled forms of *Drosophila* tubulin?”

R5.2 In our view, tube and ribbon structures represent structural intermediates for the following reasons: 1. Helical ribbons and tubes exhibit a similar pattern to microtubule growth sheets. These growing sheets represent a transient structure during microtubule assembly; they are most likely in an intermediate GTP-Pi state, and GTP γ S mimics this state. 2. We have observed the transition from Tube to MT in the negative-staining samples. Therefore, we speculate that the Tube structure represents structural intermediates of the MT assembly process and not alternative and/or misassembled forms of *Drosophila* tubulin.”

Therefore, we performed the Go model-based MD simulation results, because (1) we want to first confirm the experimentally determined “Tube-bond to MT-bond conversion” and then explore the lateral interaction formation process starting from solution tubulin structures by assigning the experimentally determined MT configuration (low free-energy state) as the potential energy minima. The simulation setup is rational considering that the MT configuration is truly the native conformation that constitutes the final state of microtubule. (2) Based on the simulation trajectories, we could further reveal key interactions between neighboring dimers during the lateral interaction formation process, which are missing in the cryo-EM structures. (3) Defining the native configuration as the targeted structure of Go model-based MD simulations to investigate the assemblies of ribosome or protein filaments³⁴⁵⁶ quite common in recent studies. (4) As replied in our first round of responses, assigning “MT-bond” state as the targeted structure does not mean the assembly process must undergo the “Tube-bond” state, and vice versa. Actually, in our simulations setting “Tube-bond” state as the targeted structure, very few MD trajectories (seven of all 310 trajectories) would undergo the “MT-bond” state before reaching the “Tube-bond” state, supporting the transition from “Tube-bond” to “MT-bond” interfaces.

As a result, we retained the simulation results. Also, we have supplemented four references in which the native configuration is defined as the targeted structure to investigate the assembly of ribosome or protein filaments.

R4.2 Just because a method is widely used, doesn't imply that it's accurate enough for the particular question at hand. Although MM/GBSA did show some agreement in a number of

studies, the authors ignore an even larger body of studies that put the accuracy of this method into question. The authors actually seem to share this view, because in their reply they resort to results by Tong & Voth as support for their MM/GBSA result. Unfortunately, the current literature also offers results that disagree with the MM/GBSA results, e.g., Igaev & Grubmuller, which the authors do cite but not here. To selectively cite literature that agrees without mentioning other literature that disagrees is not helpful. Again, this referee does not really see the point of including these simulation results within the paper, as the experiments are already sufficiently conclusive.

Response to R4.2: Thanks for the reviewer's kind suggestions. We agree with the reviewer that although the MM/GBSA method is widely used, its accuracy is relatively poor, and the MM/GBSA method was generally used for qualitative order ranking rather than quantitative calculation. In our manuscript, we mainly use the MM/GBSA method to study the order of the binding ability between the Tube-bond and MT-bond states, and the dominated residues that play major roles in different binding states. In this work, the cryo-electron microscopy (cryo-EM) experiments report a series of structures representing different MT assembly stages, which are especially two types of lateral contacts of tubulin protofilaments, namely the "Tube-bond" and "MT-bond". However the current resolution of 6.8 Å is not able to provide us more detailed structural information of these two different interfaces. Here the calculation of MM/GBSA method is used to explore the interaction of key residues around these two interfaces and qualitatively rank the binding affinity of the two states that is missing in the cryo-EM structures.

The literature of Igaev & Grubmuller mentioned by the reviewer mainly described the change of the bending torsional of MT protofilaments from the straight state to the splayed conformation under the state of binding GTP or GDP. Although the research goal of Igaev & Grubmuller is not the same as that of our manuscript, we still analyzed the intra-dimer contact map through the trajectory to make a comparison. The results show that the intra-dimer contact map is mainly contributed by the H8-helix of beta tubulin (shown below for details) and is in consistence with the work of Igaev & Grubmuller.

As a result, we retained the simulation results of MM/GBSA and made corresponding changes to compare with the work of Igaev & Grubmuller in the main text.

Figure 3 The contact map of intra-dimer interactions

The color scale indicates the relative probability of residue contact. A color change from blue to red indicates a change in probability from low to high. The helix H8 with the highest probability is labeled.

R5.1, R5.2, and R5.3 Thank you for the explanations, to which I fully agree. As above, the authors may wish to add a summary of it to the manuscript, as also other readers may wonder.

Response to R5.1, R5.2 and R5.3: Thanks for your suggestion. Now we have added a summary in the discussion section of “The structural basis for “Tube-to-MT” conversion”.

R6. I largely agree with the authors’ reply, but apparently no change to the manuscript was made.

Response to R6: Thanks for your suggestion. Now we add a brief description in the discussion section titled “Lateral interaction changes the conformation of tubulin heterodimer”.

R7. The methods part describing the MD simulations is now sufficiently detailed to render the simulations reproducible, as requested.

R8. The description of previous work has been improved satisfactorily.

References

1. Virtanen, P. *et al.* SciPy 1.0: fundamental algorithms for scientific computing in Python. *Nat Methods* **17**, 261–272 (2020).
2. Rossum, G. van & Drake, F. L. *The Python language reference*. (Python Software Foundation, 2010).
3. Nguyen, K. & Whitford, P. C. Steric interactions lead to collective tilting motion in the ribosome during mRNA–tRNA translocation. *Nat Commun* **7**, 10586 (2016).
4. Faelber, K. *et al.* Structure and assembly of the mitochondrial membrane remodelling GTPase Mgm1. *Nature* **571**, 429–433 (2019).
5. Noel, J. K., Noé, F., Daumke, O. & Mikhailov, A. S. Polymer-like Model to Study the Dynamics of Dynamin Filaments on Deformable Membrane Tubes. *Biophysical Journal* **117**, 1870–1891 (2019).
6. Levi, M., Walak, K., Wang, A., Mohanty, U. & Whitford, P. C. A steric gate controls P/E hybrid-state formation of tRNA on the ribosome. *Nature Communications* **11**, 5706 (2020).

REVIEWER COMMENTS

Reviewer #3 (Remarks to the Author):

The authors have clearly put considerable effort into addressing my previous issues, and have successfully addressed several of them. Yet some remain, as detailed below.

R1.

- The authors have addressed the problem regarding the rigid-body fitting of dimer structures into their low-resolution densities. Although the reviewer would argue that Rosetta and CDMD can handle low-resolution cases like the authors' one, their test with the low pass filtered map is convincing.

- The authors have now tried to address the previous issue of the combined angle lying outside of the respective ranges. I have to say the new, claimed improved, statistical analysis is not entirely clear to me (Fig. 1). In particular, the new 'data expansion' method did not convince me, and I do not really see how it would improve the statistics. Also, the statistical test used appears not to have used the original standard deviations, which clearly should impact the analysis. Also, looking at Fig. 2 panels I-K, one notes that panel J shows a statistically significant difference, while panel I also shows a statistically significant difference despite a substantial overlap of the densities. That adds to me being skeptical about this statistics. A simple bootstrapping of the old dataset, in my opinion, would have been a much more straightforward and mathematically solid way.

- The point with tangential vs radial flexibility has been well addressed. One minor point is that this conclusion is now based on the new improved statistical analysis that I have described above.

R2.

Although more frequent lateral encounters between protofilaments do not translate into stronger lateral bonds automatically, not having encounters at all due to reduced flexibility will not lead to bond formation either. I generally agree with the line of arguments. However, the authors do not provide data to demonstrate that GMPCPP-tubulins form lateral bonds more easily.

R3.

The point has been fully addressed.

R4.1

Here my original point has not been addressed. The authors reiterate the same arguments they used in the first round of revision, while avoiding my main point: the results they get is encoded in the Go model and thus does not provide new insights, while the experimental data is conclusive enough.

R4.2

My original point has not been addressed. The authors reiterate the same arguments they used in the first round of revision. Moreover, the authors now say that "the current resolution of 6.8 Å is not able to provide us more detailed structural information of these two different interfaces", hence admitting the experimental data may not contain the required information. The assumption that simulations + MM/GBSA (given all its numerous limitations) can provide new insights based on low-resolution data is therefore quite a bit of a stretch.

Minor concerns

All minor concerns have been fully addressed.

Reviewer #3 (Remarks to the Author):

The authors have clearly put considerable effort into addressing my previous issues, and have successfully addressed several of them. Yet some remain, as detailed below.

Response to Reviewer #3: We thank the reviewer for the good suggestions to improve our manuscript, and we have revised the manuscript accordingly.

R1. - The authors have addressed the problem regarding the rigid-body fitting of dimer structures into their low-resolution densities. Although the reviewer would argue that Rosetta and CDMD can handle low-resolution cases like the authors' one, their test with the low pass filtered map is convincing.

- The authors have now tried to address the previous issue of the combined angle lying outside of the respective ranges. I have to say the new, claimed improved, statistical analysis is not entirely clear to me (Fig. 1). In particular, the new 'data expansion' method did not convince me, and I do not really see how it would improve the statistics. Also, the statistical test used appears not to have used the original standard deviations, which clearly should impact the analysis. Also, looking at Fig. 2 panels I-K, one notes that panel J shows a statistically significant difference, while panel I also shows a statistically significant difference despite a substantial overlap of the densities. That adds to me being skeptical about this statistics. A simple bootstrapping of the old dataset, in my opinion, would have been a much more straightforward and mathematically solid way.

- The point with tangential vs radial flexibility has been well addressed. One minor point is that this conclusion is now based on the new improved statistical analysis that I have described above.

Response to R1: We appreciate the reviewer's suggestion and feedback. Based on the reviewer's suggestion, we performed bootstrapping of the original dataset, which yielded results consistent with our previous data expansion approach. In all three cases, the distribution of two datasets is significantly different. The statistical results from the three cases are summarized in Table R1. In this revised version, we generated 10,000 datasets using the bootstrapping framework from the original dataset for three cases (RB, TB, and Twist). In each newly generated dataset, we applied a two tailed T-test since the mean and standard deviation (std) values satisfied normal distributions respectively. The distributions of the two stds (GDP and GMPCPP) are significantly different in all 10,000 datasets of three cases. The distributions derived from GDP and GMPCPP were hypothetically tested, and an illustration of this process is shown in Figure R1.

Table R1. The statistical results of U-tests of RB, TB and Twist

	Number of differences (False)	Number of differences (True)
RB	0	10000
TB	0	10000
Twist	0	10000

Figure R1. The illustration of hypothetical test on generated datasets from bootstrapping
 The 10000 datasets are generated by bootstrapping the original datasets. The std of GDP and GMPCPP of newly generated 10000 datasets are hypothetically tested with T-test.

Regarding Fig. 2 panels I-K, we have revised the manuscript and figures to make them more clear. In the previous version of panel I-K in Figure 2, we utilized the two-sided Mann-Whitney U-test since the distributions of the two groups of data (GDP and GMPCPP) were not settled. In the U-test, the difference between two distributions is primarily determined by the median of each distribution¹. As shown in Figure 2I, with different medians, the two distributions exhibit a statistically significant difference, even with overlapping densities. The different medians represent different conformational states, and our focus lies in capturing the structural flexibility, which is reflected by the standard deviation (std). Therefore, we have updated Fig 2 I-J-K with violin plots displaying the std from the new dataset generated through bootstrapping and revised the description in main text.

Following the reviewer's suggestions, we believe that the revisions with clarifications and modifications strengthen the statistical analysis.

R2. Although more frequent lateral encounters between protofilaments do not translate into stronger lateral bonds automatically, not having encounters at all due to reduced flexibility will not lead to bond formation either. I generally agree with the line of arguments. However, the authors do not provide data to demonstrate that GMPCPP-tubulins form lateral bonds more easily.

Response to R2: We acknowledge that we do not have direct experimental data to demonstrate that GMPCPP-tubulins form lateral bonds more easily. However, we would like to provide further clarification and context based on our experimental data and reasoning.

Our experimental data revealed reduced structural fluctuation of GTP-tubulin compared to GDP-tubulin, indicating that GTP-tubulin may possess enhanced stability, which could potentially facilitate longitudinal or lateral bond formation. In response to Reviewer #4's suggestion, we analyzed the proportion of tubulin dimers, tetramers, and hexamers in each micrograph (Extended Data Fig. 2C). The data demonstrated that the nucleotide status did not induce a shift or increment in the tetramer/dimer ratio. This finding supports our model and suggests that the unfavorable initiation of MT assembly by GDP-tubulin is not primarily due to the impaired formation of tetramers themselves, but rather to the larger range of motion within the GDP-tubulin tetramers. Consequently, we can exclude the possibility that GMPCPP-tubulin forms longitudinal bonds more easily.

While we lack direct experimental evidence for GMPCPP-tubulins forming lateral bonds more readily, our proposed model is grounded in these observations and reasoning. The reduced structural fluctuation of GTP-tubulin and the known preference of GTP-tubulin for MT assembly provide a basis for our hypothesis that GMPCPP-tubulins, mimicking the GTP-bound state, may have an increased propensity for lateral bond formation.

In summary, although direct experimental data specifically addressing the ease of lateral bond formation by GMPCPP-tubulins are lacking, our proposed model is supported by our experimental findings and the understanding of GTP-tubulin's influence on MT assembly. We want to emphasize that this is a hypothesis offering a reasonable explanation and highlighting potential avenues for further investigation.

We thank this reviewer for raising this point to discuss. We appreciate the opportunity to provide further clarification in the main text.

R3. The point has been fully addressed.

R4.1 Here my original point has not been addressed. The authors reiterate the same arguments they used in the first round of revision, while avoiding my main point: the results they get is encoded in the Go model and thus does not provide new insights, while the experimental data is conclusive enough.

R4.2 My original point has not been addressed. The authors reiterate the same arguments they used in the first round of revision. Moreover, the authors now say that "the current resolution of 6.8 Å is not able to provide us more detailed structural information of these two different interfaces", hence admitting the experimental data may not contain the required information. The assumption that simulations + MM/GBSA (given all its numerous limitations) can provide new insights based on low-resolution data is therefore quite a bit of a stretch.

We appreciate the reviewer's comments and apologize for not directly addressing the main concerns in our previous responses. We have carefully considered the comments, and we would like to provide a combined and refined response to address these concerns:

Response to R4.1 and R4.2:

We sincerely appreciate the reviewer's thorough evaluation and valuable feedback.

In our study, we acknowledge that the resolution of our cryo-EM structures is limited to 6.8 Å, which may restrict the amount of detailed structural information that can be obtained. However, our goal was not to extract atomic-level details from the low-resolution cryo-EM structures. Instead, we aimed to use the available experimental data as a starting point and complement it with molecular dynamics (MD) simulations to gain additional insights into the underlying mechanisms.

The use of Go model-based MD simulations to investigate assemblies of ribosomes or protein filaments has become a common approach in recent studies²⁻⁵. This method has demonstrated its reliability and effectiveness in providing valuable insights into complex biological processes.

In our study, we specifically targeted the "MT-bond" and "Tube-bond" states as structures of interest within the Go model. The simulations revealed that very few MD trajectories underwent the "MT-bond" state before reaching the "Tube-bond" state, supporting our hypothesis that the Tube structure represents a structural intermediate in the microtubule assembly process. Through these simulations, we explored the detailed formation process of lateral interactions and identified key interactions between neighboring dimers that are not fully captured in the cryo-EM structures alone.

Furthermore, we performed MM/GBSA calculations to analyze the binding free energies of the "MT-bond" and "Tube-bond" interfaces. While the resolution of the cryo-EM structures may not provide detailed information about these interfaces, the simulations and MM/GBSA calculations offer insights into the dynamic behavior and energetics of the system, complementing the experimental data.

Integration of cryo-EM structures and MD simulations allows us to obtain a better understanding of the mechanism of proteins and have been widely used in the research of the structural basis of protein transport, binding etc⁶⁻¹⁰. We acknowledge the limitations of the Go model simulations and MM/GBSA calculations, as well as the challenges posed by the low-resolution data. However, these approaches have been successfully employed in various studies to provide valuable insights into protein dynamics and interactions^{11,12}.

Taking these factors into account, we firmly believe that the Go model simulations and MM/GBSA results in our study provide valuable new insights into the microtubule assembly process. They

enhance our understanding of the dynamics, energetics, and key interactions involved. Therefore, we argue that these results are necessary and should be retained in our manuscript.

We hope this combined and refined response more comprehensively addresses the concerns raised by the reviewer.

Minor concerns

All minor concerns have been fully addressed.

REFERENCE

- 1 Scheff, S. W. *Fundamental statistical principles for the neurobiologist: A survival guide*. (Academic Press, 2016).
- 2 Nguyen, K. & Whitford, P. C. Steric interactions lead to collective tilting motion in the ribosome during mRNA-tRNA translocation. *Nat Commun* **7**, 10586 (2016). <https://doi.org:10.1038/ncomms10586>
- 3 Faelber, K. *et al.* Structure and assembly of the mitochondrial membrane remodelling GTPase Mgm1. *Nature* **571**, 429-433 (2019). <https://doi.org:10.1038/s41586-019-1372-3>
- 4 Noel, J. K., Noé, F., Daumke, O. & Mikhailov, A. S. Polymer-like Model to Study the Dynamics of Dynamin Filaments on Deformable Membrane Tubes. *Biophys J* **117**, 1870-1891 (2019). <https://doi.org:10.1016/j.bpj.2019.09.042>
- 5 Levi, M., Walak, K., Wang, A., Mohanty, U. & Whitford, P. C. A steric gate controls P/E hybrid-state formation of tRNA on the ribosome. *Nat Commun* **11**, 5706 (2020). <https://doi.org:10.1038/s41467-020-19450-0>
- 6 Ahmad, K. *et al.* Structure and dynamics of an archetypal DNA nanoarchitecture revealed via cryo-EM and molecular dynamics simulations. *Nat Commun* **14**, 3630 (2023). <https://doi.org:10.1038/s41467-023-38681-5>
- 7 Mu, J. *et al.* Conformational cycle of human polyamine transporter ATP13A2. *Nat Commun* **14**, 1978 (2023). <https://doi.org:10.1038/s41467-023-37741-0>
- 8 Lancey, C. *et al.* Cryo-EM structure of human Pol κ bound to DNA and mono-ubiquitylated PCNA. *Nat Commun* **12**, 6095 (2021). <https://doi.org:10.1038/s41467-021-26251-6>
- 9 Saleh, A. *et al.* The structural basis of Cdc7-Dbf4 kinase dependent targeting and phosphorylation of the MCM2-7 double hexamer. *Nat Commun* **13**, 2915 (2022). <https://doi.org:10.1038/s41467-022-30576-1>
- 10 Yu, J. *et al.* Dynamic conformational switching underlies TFIIH function in transcription and DNA repair and impacts genetic diseases. *Nat Commun* **14**, 2758 (2023). <https://doi.org:10.1038/s41467-023-38416-6>
- 11 Travers, T., Shao, H., Wells, A. & Camacho, C. J. Modeling the assembly of the multiple domains of α -actinin-4 and its role in actin cross-linking. *Biophys J* **104**, 705-715 (2013). <https://doi.org:10.1016/j.bpj.2012.12.003>
- 12 Wu, X. *et al.* Fascin - F-actin interaction studied by molecular dynamics simulation and protein network analysis. *J Biomol Struct Dyn*, 1-10 (2023). <https://doi.org:10.1080/07391102.2023.2199083>